# REVEAL: ADVANCING RELATION-BASED VIDEO UNDERSTANDING FOR VIDEO-QUESTION-ANSWERING

## ABSTRACT

Video-Question-Answering (VideoQA) comprises the capturing of complex visual relation changes over time, remaining a challenge even for advanced Video Language Models (VLM), i.a., because of the need to represent the visual content to a reasonably sized input for those models. To address this problem, we propose RElation-based Video rEpresentAtion Learning (REVEAL) a framework designed to capture visual relation information by encoding them into structured, decomposed representations. Specifically, inspired by spatiotemporal scene graphs, we propose to encode video sequences as sets of relation triplets in the form of (*subject-predicate-object*) over time via their language embeddings. To this end, we extract explicit relations from video captions and introduce a Many-to-Many Noise Contrastive Estimation (MM-NCE) together with a Q-Former architecture to align an unordered set of video-derived queries with corresponding text-based relation descriptions. At inference, the resulting Q-former produces an efficient token representation that can serve as input to a VLM for VideoQA.

We evaluate the proposed framework on five challenging benchmarks: NeXT-QA, Intent-QA, STAR, VLEP, and TVQA. It shows that the resulting query-based video representation is able to outperform global alignment-based CLS or patch token representations and achieves competitive results against state-of-the-art models, particularly on tasks requiring temporal reasoning and relation comprehension. The code and models will be publicly released upon acceptance.

## 1 INTRODUCTION

Videos capture rich sets of information, including the static visual information of a scene and the dynamic evolution of actors, objects, and their relationships over time. Understanding these complex spatiotemporal relations poses a significant challenge for current video understanding systems, as all those aspects need to be represented efficiently. One of the main tasks in this context is the problem of VideoQA Wu et al. (2021); Xiao et al. (2021); Li et al. (2024); Yu et al. (2023). Approaches that do well here usually rely on pre-trained vision-language image backbones like CLIP Radford et al. (2021) and BLIP2 Li et al. (2023b), processing videos by extracting frame representations and combining these with Large Language Models (LLMs) Maaz et al. (2024b); Ko et al. (2023). However, these models struggle with object relations Yuksekgonul et al. (2022); Lin et al. (2024b), action detection Bansal et al. (2024); Wang et al. (2023c); Lin et al. (2024b); Momeni et al. (2023), and compositional understanding Lin et al. (2024b); Bansal et al. (2024), issues that are exacerbated with temporal sequences. While recent works have shown that LLMs can compensate those limitations via strong language priors Ko et al. (2023); Li et al. (2024); Wang et al. (2024a); Maaz et al. (2024b), image- and video-language approaches still mostly rely on global video-text alignment representations to encode the video input.

To address this problem, we propose RElation-based Video rEpresentAtion Learning (REVEAL). This framework learns video representations by explicitly modeling the content as object relations over time via relation triplets in the form of (*subject-predicate-object*). Our relation-based approach is inspired by prior work from video scene graphs context Cong et al. (2021); Ji et al. (2020); Rodin et al. (2024); Urooj et al. (2023). However, scene graphs usually encode triplets via class indices, limiting the setting to close-ended and hand-annotated small-scale scenarios and hindering scalability. Inspired by this, REVEAL seeks to leverage this representation to learn general open-ended and web-supervised representations for video data.

To achieve this, we first leverage LLMs to convert captions into one or more relation triplets, allowing us to source triplets at scale. The resulting triplets can be considered minimum viable sentences, allowing a standard text encoding, *e.g.*, by a sentence encoder, resulting in one embedding representation per triplet and $J$ relation embeddings to describe a particular video. On the video side, we leverage a Q-Former architecture Carion et al. (2020) to encode the visual representation of one or more frames into a fixed set of vision queries. To train the Q-Former, we must match the fixed number of unordered vision queries to a variable number of unordered text triplet representations. To address this problem, we propose a Many-to-Many Noise Contrastive Estimation (MM-NCE) loss formulation, which aligns two sets of matching but unordered, incomplete sets, *e.g.*, in our case, vision-based queries with corresponding text-based relation embeddings. Practically, MM-NCE maximizes the similarity between matched query-relation pairs while contrasting them against all unmatched pairs. This allows us to train the Q-former so that the resulting query tokens approximate the relation encodings of the video. The resulting vision queries can then be used to fine-tune a standard VLM architecture to address video-language-related tasks such as VideoQA.

We evaluate REVEAL on five VideoQA datasets, NeXT-QA, Intent-QA, STAR, VLEP, and TVQA, demonstrating competitive performance compared to state-of-the-art methods. It shows that query-based representations, empowered by MM-NCE, are particularly effective at connecting video and text when adapting to LLMs. Our analysis further reveals that initializing the relation encoder with a contrastively trained sentence embedder significantly enhances semantic alignment compared to alternatives like CLIP's text encoder. We summarize the contributions of this work as follows:

- We propose a new encoding for web-based video learning by modeling relations in videos as target representation.

- We propose a MM-NCE loss for contrastive learning over two sets of matching but unordered, incomplete sets.

- We provide an extensive evaluation showing the efficacy of query-based representations and the role of MM-NCE in the context of state-of-the-art VideoQA architectures.

## 2 RELATED WORK

**VLMs for VideoQA** Video understanding, particularly VideoQA, has witnessed significant advancement with the emergence of LLMs and Large Vision-Language Models. Early approaches to VideoQA emerged in response to increasingly challenging benchmarks designed to test various aspects of video understanding. The complexity of VideoQA as a task is evidenced by the diverse set of benchmarks, each targeting different reasoning capabilities: TVQA Lei et al. (2018) challenged models with understanding TV show content requiring integration of visual cues and dialogue; STAR Wu et al. (2021) focused on situated reasoning about object interactions in indoor environments; NeXT-QA Xiao et al. (2021) emphasized causal and temporal reasoning across everyday activities; Intent-QA Li et al. (2023a) specifically tested models' ability to understand human intentions and motivations behind observed actions; and VLEP Lei et al. (2020) evaluated models' capacity to predict future events based on observed video content.

In addressing these challenges, early approaches predominantly treated VideoQA as a classification task, where video and question features were fed into classification layers to select from a fixed set of answer choices Jang et al. (2017b); Xu et al. (2017); Fan et al. (2019). These methods typically employed CNN-RNN architectures, attention mechanisms, or memory networks to capture temporal dynamics, but their classification-based paradigm fundamentally limited their reasoning capabilities and prevented them from leveraging the generative power and world knowledge inherent in modern LLMs. Graph-based approaches like SHG-VQA Urooj et al. (2023) and VGT Xiao et al. (2022) attempted to model explicit relations between objects but remained constrained by closed-vocabulary limitations, small-scale datasets, and the classification-based framework. These methods struggled with reasoning tasks due to a lack of semantic understanding.

Recent approaches have explored the direct application of VLMs to videos. IG-VLM Kim et al. (2024) represents videos as image grids, while SLOWFAST-LLaVA Xu et al. (2024) employs multi-scale temporal pooling for feature extraction. While effective for general understanding, these methods often struggle with complex temporal reasoning, which REVEAL addresses through explicit relation modeling. Further, the success of instruction-tuning in image-LLM connections Hu et al. (2022); Li et al. (2024); Beyer et al. (2024); Liu et al. (2024); Xiao et al. (2024) has inspired sim-

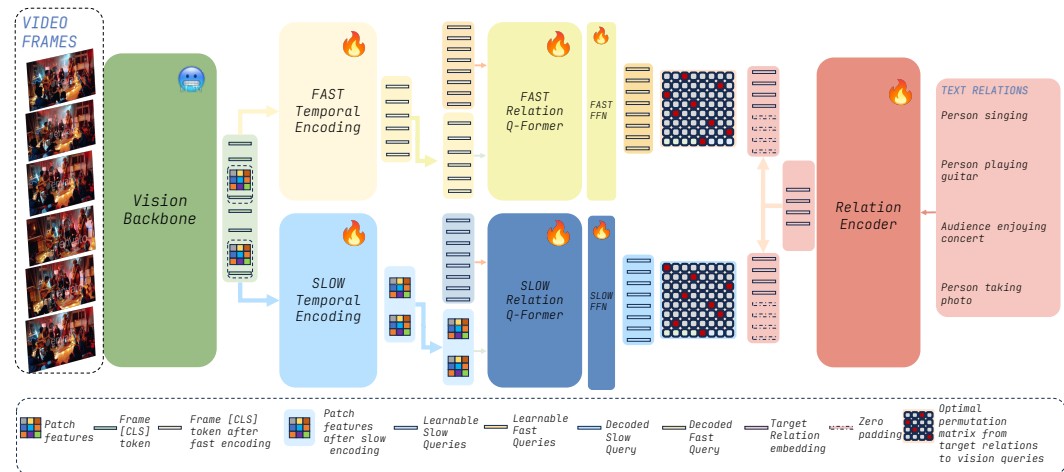

Figure 1: Overview of REVEAL: dual-pathway (Fast/Slow) vision encoders, temporal encoders, Relation Q-Formers, and a sentence-based Relation Encoder aligned via MM-NCE.

ilar approaches for video understanding. Video-ChatGPT Maaz et al. (2024b), VideoChat Li et al. (2023c), and their successors VideoChat2 Li et al. (2024) and VideoGPT+ Maaz et al. (2024a) focus on video-conversation capabilities. Notable advances include Video-Llama Zhang et al. (2023)'s multi-modal processing, Video-LLaVA Lin et al. (2024a)'s unified representation space, and MotionEpic Fei et al. (2024)'s "Video-of-Thought" framework. Llama-VQA Ko et al. (2023) and Vamos Wang et al. (2023b) finetune adapters specifically for VideoQA. LLaVA-Next-Interleave Li et al. (2025b), MPLUG-OWL-3 Ye et al. (2025), and LLaVA-One Vision Li et al. (2025a) have further advanced instruction-tuning approaches with powerful vision backbones. Finally, recent works have focused on unsupervised frame selection for this task, like Sevila, Vila, and LVNet Yu et al. (2024); Wang et al. (2024b); Park et al. (2024). These approaches use large vision backbones and Gumbel-Softmax Jang et al. (2017a) to discriminate frames, achieving strong results with a handful of frames. While orthogonal to REVEAL's relation-based approach, future work could combine these methods for more efficient videoQA.

**Video-Language Pretraining** Video-language pretraining has evolved significantly, with diverse architectural paradigms emerging to address the challenges of temporal modeling and multimodal alignment. Several key approaches have shaped this landscape: Q-former-based architectures like BLIP-2 Junnan et al. (2022) and its video adaptations Hang et al. (2023); Lin et al. (2024a) use query-based cross-attention to bridge vision and language models; encoder-decoder frameworks like InternVideo Wang et al. (2022) and InternVideo2 Wang et al. (2024c) combine masked video modeling with video-language contrastive learning; and unified architectures such as All-in-One Alex et al. (2022) employ "token rolling" for efficient temporal modeling. Contrastive learning approaches have been particularly influential, with works like FrozenBiLM Bain et al. (2021), VideoCLIP Hu et al. (2021), CLIP4Clip Huaishao et al. (2021) and CLIP2Video Han et al. (2021), establishing effective video-text alignment techniques. Temporal modeling has been addressed through hierarchical approaches in HiTeA Qinghao et al. (2022) and HERO Linjie et al. (2020), while UniVL Huaishao et al. (2020) pioneered joint understanding and generation objectives. Recent advances include VidL Cheng et al. (2023), which presents a progressive recipe for video-language model construction, and MERLOT Zellers et al. (2022), which leverages YouTube transcripts for self-supervised learning.

## 3 RELATION-BASED VIDEO REPRESENTATION LEARNING (REVEAL)

REVEAL is a framework designed to capture visual relation information in videos by encoding them into structured, decomposed representations. This section is structured as follows: Sec. 3.1 describes the relation triplet sourcing from video captions, Sec. 3.2 the overall architecture of REVEAL, Sec. 3.3 details the relation modeling, Sec. 3.4 describes the MM-NCE loss for aligning unordered sets of relations, and Sec. 3.5 covers the implementation details.

### 3.1 RELATION EXTRACTION FROM VIDEO CAPTIONS

We develop a relation extraction pipeline to transform natural language video captions into structured relation triplets (*subject-predicate-object*). Traditional approaches often depend on manually annotated datasets or rule-based methods, limiting scalability Thomee et al. (2016); Yu et al. (2023); Shang et al. (2019); Sigurdsson et al. (2016). Compared to that, REVEAL leverages the Mistral-7B model Jiang et al. (2023) to automate and scale extraction from large-scale datasets like WebVid-2M Bain et al. (2021).

To guide the LLM in decomposing unstructured captions into meaningful relation triplets, we use in-context learning, detailed in Supplement Sec. E, to identify and extract relevant relations. This pipeline automatically generates multiple relation triplets per video, as illustrated in Figure 2, providing a decomposed representation of the video caption with respect to the visual content.

### 3.2 REVEAL ARCHITECTURE

Our approach represents videos as sets of relation triplets in the form of *subject-predicate-object*. Unlike methods relying on finite indexed triplets Urooj et al. (2023) or separate object-predicate classification Herzig et al. (2023); Salzmann et al. (2025), we learn relation representations from language embeddings by aligning video-derived queries with text-derived relation embeddings.

As shown in Figure 1, the REVEAL architecture consists of four main components: (1) a vision encoder to compute frame-level features via a pretrained backbone; (2) a temporal encoder to capture the temporal dependencies across features from different frames; (3) a Relation Q-Former to transform the resulting visual features into vision queries; and (4) a Relation Encoder to encode text-based relation triplets for supervision. During training, our MM-NCE loss aligns the vision queries with relation embeddings through Hungarian matching, optimizing all components except the frozen vision backbone.

### 3.3 RELATION MODELING

For a video $\mathcal{V}$, we begin with transforming the video into visual tokens. A visual encoder $f(.)$ processes each video's frames, producing a set of features: $(\mathbf{x}_n)_{n \in \{1..N\}} = f(\mathcal{V})$, where $N$ denotes the number of tokens per video. These tokens serve as input for relation modeling.

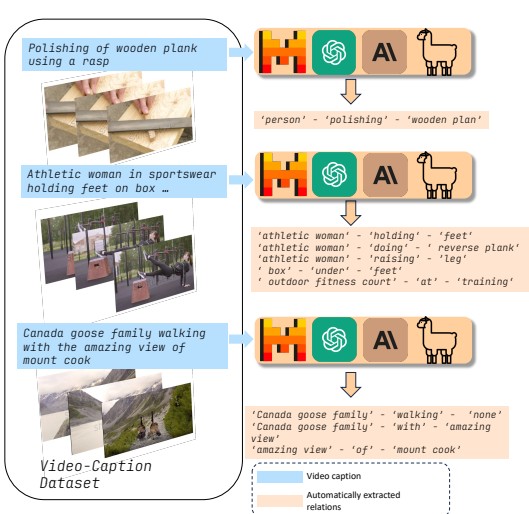

**Relation Q-former**: To transform learnable queries $(\mathbf{v}_m^0)_{m \in \{1..M\}}$ into vision queries $(\mathbf{v}_m)_{m \in \{1..M\}}$, we employ a Q-former architecture Carion et al. (2020). This module performs cross-attention between the initial queries and the video's visual tokens $(\mathbf{x}_n)_{n \in \{1..N\}}$: $(\mathbf{v}_m)_{m \in \{1..M\}} = g((\mathbf{v}_m^0)_{m \in \{1..M\}}, (\mathbf{x}_n)_{n \in \{1..N\}})$, The resulting queries are processed through a feed-forward network to yield relation embeddings aligned with text-derived triplets.

**Relation Encoder**: In parallel, text relations $(\mathbf{t}_j)_{j \in \{1..J\}}$ associated with the video are passed through a text encoder $h(.)$ to get relation embeddings $(\mathbf{r}_j)_{j \in \{1..J\}} = h((\mathbf{t}_j)_{j \in \{1..J\}})$. Practically, we leverage a pre-trained sentence embedder, initialized with contrastively trained models like Sentence-BERT Reimers & Gurevych (2019).

Figure 2: Relation extraction pipeline: Mistral-7B decomposes WebVid-2M captions into (subject-predicate-object) triplets.

Finally, the vision queries $(\mathbf{v}_m)_{m \in \{1..M\}}$ are aligned with the text-derived relation embeddings $(\mathbf{r}_j)_{j \in \{1..J\}}$ via the proposed MM-NCE loss.

### 3.4 RELATION LOSS FUNCTION: MANY-TO-MANY NOISE CONTRASTIVE ESTIMATION

We introduce Many-to-Many Noise Contrastive Estimation (MM-NCE) as a contrastive learning approach designed to align unordered sets of relations. The key challenge is that relation triplets extracted from video captions form an unordered set with no predefined temporal correspondence to visual elements in the video. This presents two difficulties: the number of extracted relation triplets may differ from the number of visual queries, requiring a flexible matching strategy, and unlike traditional video-text alignment where a single caption corresponds to an entire video, our approach must determine which specific vision query correspond to which relation embedding without explicit supervision. For a batch with samples $k \in \mathcal{B}$, we consider each video as $\mathcal{V}_k$, the text-derived relation embeddings as $(\mathbf{r}_j^{(k)})_{j \in \mathcal{J}^{(k)}}$, with $\mathcal{J}^{(k)} = \{1..J^{(k)}\}$, and the vision queries as $(\mathbf{v}_m^{(k)})_{m \in \{1..M\}}$. We first determine the optimal matching between them using Hungarian matching, therefore creating a set of query-relation positive pairs for each video in the batch:

$$\sigma^{(k)} = \mathrm{argmax}_{\sigma \in \mathcal{S}_{J^{(k)},M}} \sum_{j \in \mathcal{J}^{(k)}} s_c\left(\mathbf{r}_j^{(k)}, \mathbf{v}_{\sigma(j)}^{(k)}\right), \tag{1}$$

where $\mathcal{S}_{J^{(k)},M}$ is the set of injective mappings from $\mathcal{J}^{(k)}$ to $\{1..M\}$ and with the cosine similarity

$$s_c(\mathbf{r}, \mathbf{v}) = \frac{\mathbf{r}^T \mathbf{v}}{\|\mathbf{r}\|\|\mathbf{v}\|}. \tag{2}$$

Note that, in equation 1, not all vision queries are paired to a text-derived relation embedding when $J^{(k)} < M$; the resulting mapping $\sigma^{(k)}(.)$ is injective but not surjective. This is a key property of our approach: it is designed to handle varying numbers of text relations per video. Eventually, only paired vision queries contribute to the loss defined below.

In the following equations, we omit the learnable parameters. The MM-NCE loss then consists of two symmetric terms. $L_q$ measures query-to-relation alignment:

$$L_{q \to r} = - \sum_{\substack{k \in \mathcal{B} \\ j \in \mathcal{J}^{(k)}}} \log \frac{\exp\left(s_c\left(\mathbf{r}_j^{(k)}, \mathbf{v}_{\sigma^{(k)}(j)}^{(k)}\right)/\tau\right)}{\sum_{\substack{k' \in \mathcal{B} \\ i \in \mathcal{J}^{(k')}}} \exp\left(s_c\left(\mathbf{r}_i^{(k')}, \mathbf{v}_{\sigma^{(k)}(j)}^{(k)}\right)/\tau\right)}, \tag{3}$$

and for the relation-to-query alignment term $L_{r \to q}$:

$$L_{r \to q} = - \sum_{\substack{k \in \mathcal{B} \\ j \in \mathcal{J}^{(k)}}} \log \frac{\exp\left(s_c\left(\mathbf{r}_j^{(k)}, \mathbf{v}_{\sigma^{(k)}(j)}^{(k)}\right)/\tau\right)}{\sum_{\substack{k' \in \mathcal{B} \\ i \in \{1..M\}}} \exp\left(s_c\left(\mathbf{r}_j^{(k)}, \mathbf{v}_i^{(k')}\right)/\tau\right)}. \tag{4}$$

Here, $k'$ and $i$ index over all videos in batch $\mathcal{B}$ and vision queries from a video, respectively, creating negative pairs from other videos. The temperature parameter $\tau$ is learnable.

$L_{q \to r}$ and $L_{r \to q}$ allow us to compute the MM-NCE-loss:

$$L_{\text{MM-NCE}} = L_{q \to r} + L_{r \to q}. \tag{5}$$

MM-NCE pulls matched query-relation pairs closer in embedding space while pushing negative pairs apart, handling the unordered nature of relations through Hungarian matching rather than requiring predefined correspondences. It specifically allows the handling of varying numbers of annotations per video. When some vision queries are not matched to annotated relations, the model can freely learn to model relations in a video even when not annotated if they appear in other videos in the training data. Thus, it can also deal with non-exhaustive annotation. Unlike Multiple Instance Learning and Noise Contrastive Estimation (MIL-NCE) Miech et al. (2020), designed to align multiple captions to a single representation, this approach enforces a one-to-one correspondence between the multiple video representations, *i.e.*, the vision queries, and the corresponding text relations.

### 3.5 IMPLEMENTATION DETAILS

**Slow-Fast Video Processing**   Following recent work Xu et al. (2024); Maaz et al. (2024a), RE-VEAL employs a dual-pathway architecture to capture both global context and fine-grained spatial information, enhancing relation understanding while balancing computational efficiency: the **Fast Pathway** uses [CLS] tokens across 16 frames for efficient temporal aggregation and high-level motion understanding; the **Slow Pathway** processes patch features from four carefully selected frames for detailed spatial information and object-level relationship modeling. Each pathway processes its respective features using a dedicated temporal encoder and relation Q-former.

The temporal encoders model dependencies across frames, with the Fast pathway capturing global changes and long-range temporal dynamics and the Slow pathway specifically modeling patch relationships across frames for fine-grained spatial reasoning. The relation Q-Formers perform cross-attention with visual features to transform learnable queries into relation embeddings representing meaningful subject-predicate-object triplets.

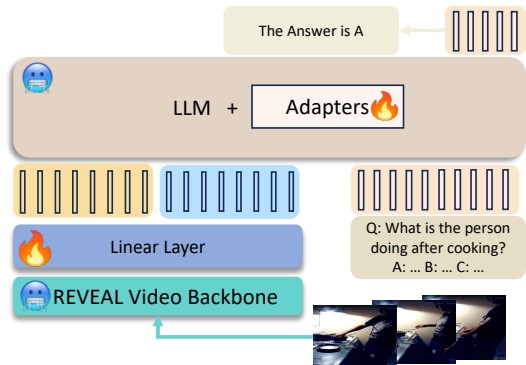

Figure 3: Overview of the VideoQA finetuning approach. The framework integrates pre-trained relation embeddings from our model with LLMs via adapters.

**VideoQA Finetuning**   We evaluate RE-VEAL on multiple-choice VideoQA by adapting frozen video–relation features to LLMs via Llama adapters Zhang et al. (2024), as in Figure 3 and prior work Wang et al. (2023b); Ko et al. (2023). Videos are split into 1–8 segments; each segment yields 16 vision queries (8 per pathway), giving 16–128 embeddings per video, which are linearly projected into the LLM space. For temporal alignment, each 16-query group maps to its segment, with special tokens marking Slow vs. Fast outputs and learnable temporal tokens encoding segment positions. Training follows Flipped-VQA Ko et al. (2023): the main task VQ→A (answers from video queries + question) and two auxiliaries, VA→Q and QA→V, to reduce linguistic bias and improve visual grounding; REVEAL remains frozen.

| Method | Specifications | Language Backbone | Vision Backbone | Int | Seq | Pred | Feas | All |
|---|---|---|---|---|---|---|---|---|
| SHG-VQA (val set) Urooj et al. (2023) | FT | BERT | SlowR50-K400 | 48.0 | 42.0 | 35.3 | 32.5 | 39.5 |
| All-in-One Wang et al. (2023a) | PT + FT | All-in-One | All-in-One | 47.5 | 50.8 | 47.7 | 44.0 | 47.5 |
| InternVideo Wang et al. (2022) | PT + FT | CLIP text encoder | ViT-H/14 | 62.7 | 65.6 | 54.9 | 51.9 | 58.7 |
| Sevila Yu et al. (2024) | FT + FS | BLIP-2 (FlanT5-XL) | BLIP-2 (ViT-G/14) | 63.7 | **70.4** | 63.1 | **62.4** | 64.9 |
| ViLA Wang et al. (2024b) | FT + FS | BLIP-2 (FlanT5-XL) | BLIP-2 (ViT-G/14) | **70.0** | **70.4** | 65.9 | 62.2 | **67.1** |
| IG-VLM Kim et al. (2024) | ZS | Llava 1.6 | ViT-L/14 | 49.3 | 50.1 | 49.5 | 48.8 | 49.6 |
| Llama-VQA Ko et al. (2023) (baseline) | LLM-A | Llama1 | ViT-L/14 | 66.2 | 67.9 | 57.2 | 52.7 | 65.4 |
| REVEAL (ours) | PT + LLM-A | Llama1 | ViT-L/14 | 60.0 | 70.7 | **72.5** | 68.4 | 67.9 |
| Llama-VQA* Ko et al. (2023) (baseline) | LLM-A | Llama3 | ViT-L/14 | **59.8** | 67.2 | 59.8 | 50.4 | 65.4 |
| REVEAL (ours) | PT + LLM-A | Llama3 | ViT-L/14 | 59.7 | 70.8 | 70.7 | 68.7 | 67.5 |

Table 1: STAR results. Specs: PT=Pretrain, FT=Finetune, FS=FrameSel, ZS=Zero-Shot, LLM-A=LLM+Adapters. *=our run.

## 4 EXPERIMENTS

### 4.1 DATASETS

**Pretraining Datasets:**   We pretrain REVEAL on the **WebVid-2M** dataset, a large-scale collection of **2.5 million** video-caption pairs sourced from public web platforms Bain et al. (2021). Relation triplets are extracted from captions using the **Mistral-7B** model Jiang et al. (2023). Post-extraction, automated filtering removes ambiguous or redundant triplets, yielding an average of four relations per video. To enhance relation diversity and robustness, we incorporate annotations of 8k videos from **Charades** Sigurdsson et al. (2016) and 3k videos from **VidOR** Shang et al. (2019), splitting each video into clips with 4–8 relations, adding approximately 80k clips to the training set. To prevent data leakage, we ensure no selected clips from Charades or VidOR overlap with STAR, NeXT-QA, or Intent-QA evaluation sets. The relation extraction and filtering details are provided in the supplement section E.

| Method | Specifications | Language Backbone | Vision Backbone | Caus | Temp | Des | All |
|---|---|---|---|---|---|---|---|
| All-in-One Wang et al. (2023a) | PT + FT | All-in-One | All-in-One | 48.6 | 48.0 | 63.2 | 50.6 |
| Video-Llama Zhang et al. (2023) | IT + ZS | Llama | ViT-G/14 | 57.4 | 59.2 | 72.3 | 60.6 |
| VideoChat Li et al. (2024) | IT + ZS | StableVicuna | BLIP-2 (ViT-G/G) | 61.5 | 63.5 | 82.1 | 61.8 |
| HiTeA Ye et al. (2023) | PT + FT | BERT-Base | MViT-Base | 58.3 | 62.4 | 75.6 | 63.1 |
| InternVideo Wang et al. (2022) | PT + FT | CLIP text encoder | ViT-H | 58.5 | 62.5 | 75.8 | 63.2 |
| VideoChat2 Li et al. (2024) | IT + ZS | Llama1 | UMT-L | 64.7 | 68.7 | 76.1 | 68.6 |
| LVNet Fei et al. (2024) | ZS + FS | GPT-4o | GPT-4o | 65.5 | **75.0** | **81.5** | 72.9 |
| Sevila Yu et al. (2024) | FT + FS | BLIP-2 (FlanT5-XL) | BLIP-2 (ViT-G/14) | 69.4 | 74.4 | 81.3 | 73.8 |
| ViLA Wang et al. (2024b) | FT + FS | BLIP-2 (FlanT5-XL) | BLIP-2 (ViT-G/14) | **71.4** | 73.6 | 81.4 | **74.8** |
| IG-VLM Kim et al. (2024) | VLM + ZS | LLava 1.6 | ViT-L/14 | 63.1 | 57.3 | 74.9 | 63.1 |
| SLOWFAST-LLava Xu et al. (2024) | VLM + ZS | LLava-Next | ViT-L/14 | – | – | – | 64.2 |
| Video-ChatGPT Maaz et al. (2024b) | IT + ZS | LLaVA | ViT-L/14 | 64.1 | 66.9 | 75.7 | 64.4 |
| Flipped-VQA (baseline) Kim et al. (2024) | LLM + A | Llama1 | ViT-L/14 | 72.7 | 69.2 | 75.8 | 72.0 |
| REVEAL (ours) | PT + LLM-A | Llama1 | ViT-L/14 | 73.7 | 69.2 | 76.5 | 72.7 |
| REVEAL (ours) | PT + LLM-A | Llama3 | ViT-L/14 | **75.3** | **69.9** | **78.5** | **74.0** |
| Vamos Wang et al. (2023b)* | C + LLM-A | Llama3 | ViT-L/14 | 76.1 | 73.7 | 80.4 | 76.0 |
| REVEAL (ours) | C + PT + LLM-A | Llama3 | ViT-L/14 | **77.8** | 74.4 | **81.9** | 77.2 |
| Vamos Wang et al. (2023b) | C + LLM-A | Llama3 | ViT-L/14 | 77.2 | 75.3 | 81.7 | 77.3 |
| LLaVA-Next-Interleave Li et al. (2025b) | IT + ZS | QWEN-1.5 | SigLIP | – | – | – | 77.9 |
| MPLUG-OWL-3 Ye et al. (2025) | IT | QWEN-2 | SigLIP | – | – | – | 78.6 |
| LLaVA-One Vision Li et al. (2025a) | IT | QWEN-2 | SigLIP | – | – | – | **79.4** |

Table 2: NExT-QA results (causal, temporal, descriptive). * indicates reproduced results.

**VideoQA Evaluation Datasets**: **VideoQA Evaluation Datasets**: We finetune and evaluate on five benchmarks: ◇**STAR** Wu et al. (2021): 60K questions over 22K indoor clips across interaction, sequence, prediction, and feasibility; probes object interactions and consequences. ◇**NExT-QA** Xiao et al. (2021): 52K QA pairs over 5,440 videos; causal (48%), temporal (29%), descriptive (23%). ◇**Intent-QA** Li et al. (2023a): extends NExT-QA with 16K QA pairs on intention understanding across four types. ◇**TVQA** Lei et al. (2018): 152K QA pairs over 21K TV show clips (5-choice); dialogue-heavy. ◇**VLEP** Lei et al. (2020): binary next-event prediction with 28K examples over 10K clips. We report answer accuracy, with category breakdowns for STAR, NExT-QA, and Intent-QA.

| Method | Specifications | Language Backbone | Vision Backbone | CW | CH | TP&TN | All |
|---|---|---|---|---|---|---|---|
| HQGA Le et al. (2021) | FT | BERT | ResNeXt-101/ResNet-101 | 48.2 | 54.3 | 41.7 | 47.7 |
| VGT Xiao et al. (2022) | FT | BERT | VGT | 51.4 | 55.9 | 47.6 | 51.3 |
| CaVIR Li et al. (2023a) | FT | BERT | VGT | 58.4 | 65.4 | 50.5 | 57.6 |
| VideoChat Lin et al. (2024a) | IT + ZS | StableVicuna | BLIP-2 (ViT-G/14) | – | – | – | 59.3 |
| LVNet Fei et al. (2024) | ZS + FS | GPT-4o | GPT-4o | 75.0 | 74.4 | 62.1 | 71.7 |
| Video-LLaVA Lin et al. (2024a) | IT + ZS | Vicuna-7B | ViT-L/14 | – | – | – | 62.5 |
| Flipped-VQA* Ko et al. (2023) | LLM-A | LLama3 | ViT-L/14 | 73.7 | 72.6 | 57.3 | 69.5 |
| REVEAL (ours) | PT + LLM-A | Llama3 | ViT-L/14 | **74.0** | **77.4** | **66.8** | **72.8** |
| Vamos Wang et al. (2023b) | C + LLM-A | Llama3 | ViT-L/14 | 75.1 | 77.4 | 69.5 | 74.1 |
| REVEAL (ours) | PT + C + LLM-A | Llama3 | ViT-L/14 | **77.9** | 77.3 | 67.5 | **75.0** |

Table 3: Performance comparison on Intent-QA dataset for intention understanding through causal and temporal reasoning (CW: Causal Why, CH: Causal How, TP&TN: Temporal Previous & Next). * indicates that we run the baseline evaluation ourselves.

## 4.2 TRAINING DETAILS

Using CLIP's ViT-L/14 Radford et al. (2021) as the vision backbone, we process frame features with a two-layer transformer encoder followed by a 12-layer Q-Former module for both the slow and the fast pathway output, resulting in eight learnable queries per pathway. This yields 16 vision query tokens per video clip, which are projected into the relation embedding space via a fully connected feed-forward network. The relation encoder is initialized with a pretrained sentence embedder ("all-roberta-large-v1" from Thakur et al. (2021)) based on Sentence-BERT Reimers & Gurevych (2019) and the RoBERTa-large architecture Liu et al. (2019). It transforms relation triplets, formatted as "Subject: *subj*, Predicate: *pred*, Object: *obj*", into single 1024-dimensional embeddings. Depending on the caption, one video can have multiple associated triplets. If more than eight text-derived relation embeddings are available, we randomly sample eight triplets. The resulting embedding sequence is further adapted with a one-layer feed-forward network.

We pretrain the model for five epochs on eight MI210 GPUs for approximately one day with the AdamW optimizer and a cosine-decayed learning rate of $5 \times 10^{-5}$. The resulting model comprises 590 million parameters.

We finetune the model for each benchmark separately. To this end, we follow best practices of previous works Ko et al. (2023); Wang et al. (2023b), keeping our pretrained video model, REVEAL,

frozen and finetuning only a Linear layer and the Llama backbone via Llama-adapters Zhang et al. (2024) considering both Llama1 (7B) and Llama3 (8B)Touvron et al. (2023); Dubey et al. (2024) as our language models.

## 4.3 COMPARISON TO STATE-OF-THE-ART METHODS

**STAR:** Table 1 shows the comparison with state-of-the-art approaches on STAR. We improve by 2.5% compared to the Flipped-VQA baseline Ko et al. (2023), with the same vision, language backbones, and finetuning setting. Furthermore, we achieve state-of-the-art results improving upon ViT-G/14-based ViLA Wang et al. (2024b) by 0.8% while using the significantly less powerful ViT-L/14. The most substantial gains appear in prediction (+6.6%) and feasibility (+6.2%) questions testing the understanding of interactions and temporal reasoning.

**NExT-QA:** On NExT-QA (Table 2), REVEAL with Llama3 achieves 74.0% accuracy and 72.7% with Llama1, outperforming the Flipped-VQA baseline (72.0%) using identical vision backbones. Additionally, we implement a REVEAL+Captioning baseline for comparison with Vamos Wang et al. (2023b), integrating off-the-shelf captioning to complement relation embeddings with text descriptions. This improves performance to 77.2% and is on par with Vamos's results (77.3%) while outperforming our reproduced baseline by 1.2%.

**Intent-QA:** Table 3 shows REVEAL achieving 72.8% accuracy on Intent-QA, beating the Flipped-VQA baseline 3.3%. With complementary captions, REVEAL reaches 75.0%, surpassing Vamos (74.1%), with identical backbones.

**TVQA and VLEP Datasets:** On TVQA (Table 4), REVEAL achieves state-of-the-art performance (83.0%), outperforming Flipped-VQA (82.2%) by 0.8%. Similarly, on VLEP (Table 5), we achieve 73.5% surpassing Flipped-VQA by 2.5% and 1.2% with Llama1 and Llama3, respectively.

| Method | Specs. | Language | Vision | All |
|---|---|---|---|---|
| InternVid Wang et al. (2022) | PT+FT | CLIP | ViT-H | 57.2 |
| Merlot Zellers et al. (2022) | PT+FT | RoBERTa | ResNet-50 | 78.7 |
| VidL Cheng et al. (2023) | PT+FT | BERT | ViT-B/16 | **79.0** |
| FrozenBiLM Bain et al. (2021) | PT+ZS | DeBERTa | ViT-L/14 | 82.0 |
| Flipped-VQA Kim et al. (2024) | LLM-A | Llama1 | ViT-L/14 | 82.2 |
| REVEAL | PT+LLM-A | Llama3 | ViT-L/14 | **83.0** |

Table 4: Performance on TVQA dataset.

| Method | Specs. | Language | Vision | All |
|---|---|---|---|---|
| InternVideo Wang et al. (2022) | PT+FT | CLIP | ViT-H | 63.9 |
| Merlot Zellers et al. (2022) | PT+FT | RoBERTa | ResNet-50 | **68.4** |
| VideoChat Li et al. (2024) | IT+ZS | StableVicuna | ViT-G/14 | 62.0 |
| SeViLA Yu et al. (2024) | FT+FS | FlanT5-XL | ViT-G/14 | 68.9 |
| ViLA Wang et al. (2024b) | FT+FS | FlanT5-XL | ViT-G/14 | **69.6** |
| Video-LLaVA Lin et al. (2024a) | IT+ZS | Vicuna-7B | ViT-L/14 | 65.8 |
| Flipped-VQA Kim et al. (2024) | LLM-A | Llama1 | ViT-L/14 | 71.0 |
| Flipped-VQA* Kim et al. (2024) | LLM-A | Llama3 | ViT-L/14 | 72.3 |
| REVEAL | PT+LLM-A | Llama3 | ViT-L/14 | **73.5** |

Table 5: Performance on VLEP dataset. * indicates that we run the baseline evaluation ourselves.

## 4.4 ABLATION STUDIES

We conduct ablations to validate REVEAL's key components, summarized in Tables 6 and 7 across STAR, NExT-QA, and Intent-QA.

**Video-Relation vs. Video-Caption Alignment**: Table 6.a compares relation modeling with MM-NCE to caption-based NCE. Relations+MM-NCE yield substantial gains over captions, confirming that structured triplets outperform captions.

| Ablation | STAR | | | | | NeXT-QA | | | | Intent-QA | | | |
|---|---|---|---|---|---|---|---|---|---|---|---|---|---|
| | In | Seq | Pre | Feas | All | C | T | D | All | CW | CH | TN | All |
| *a) Annotations:* | | | | | | | | | | | | | |
| Captions + NCE loss | 32.1 | 35.2 | 28.7 | 29.6 | 31.5 | 58.4 | 56.1 | 49.7 | 56.3 | 69.2 | 63.5 | 50.8 | 61.2 |
| relations + MM-NCE loss | **58.4** | **65.6** | **69.1** | **68.4** | **65.4** | **74.0** | **68.3** | **77.7** | **72.8** | **74.9** | **74.0** | **62.5** | **70.8** |
| *b) Rel. Enc.:* | | | | | | | | | | | | | |
| Frozen + MSE loss | 59.3 | 68.9 | 75.2 | 70.6 | 66.4 | 73.1 | 66.6 | 73.5 | 71.1 | 73.9 | 73.9 | 55.5 | 68.9 |
| Frozen + MM-NCE loss | 59.7 | 69.0 | 73.1 | 69.8 | 67.9 | 73.8 | 68.8 | 76.2 | 72.6 | 73.7 | 74.4 | 63.1 | 71.4 |
| Trainable + MM-NCE loss | **61.4** | **69.3** | **75.0** | **72.0** | **69.4** | **75.3** | **69.9** | **78.5** | **74.0** | **74.6** | **75.5** | **65.6** | **71.8** |
| *c) LLM's video input:* | | | | | | | | | | | | | |
| Without FFN layer | 54.6 | 61.0 | 64.7 | 67.8 | 62.0 | 73.3 | 68.1 | 76.3 | 72.1 | 72.8 | 74.3 | 56.9 | 70.0 |
| With FFN layer | **61.4** | **69.3** | **75.0** | **72.0** | **69.4** | **75.3** | **69.9** | **78.5** | **74.0** | **74.6** | **75.5** | **65.6** | **71.8** |
| *d) Pathways:* | | | | | | | | | | | | | |
| Slow | **62.1** | 68.9 | 74.2 | 70.2 | 68.9 | 73.0 | 68.2 | 76.1 | 71.9 | 74.0 | 73.3 | 60.6 | 70.3 |
| Fast | 57.5 | 65.5 | 68.1 | 69.6 | 65.2 | 73.7 | 68.4 | 77.5 | 72.6 | 73.1 | 74.2 | 66.3 | 71.1 |
| Slow-Fast | 61.4 | **69.3** | **75.0** | **72.0** | **69.4** | **75.3** | **69.9** | **78.5** | **74.0** | **74.6** | **75.5** | 65.6 | **71.8** |

Table 6: a) Pretraining on relations compared to training on captions. Both models were pre-trained on WebVid only. The caption model was contrastively trained by attention pooling on the vision queries. b) Ablation on the trainable relation encoder c) Results of using the vision queries compared to the last hidden states from REVEAL. d) Ablation on the slow-fast architecture.

**Trainable vs. Frozen Relation Encoder**: Table 6.b compares (i) Hungarian matching + MSE and (ii) the proposed MM-NCE, both with a frozen sentence embedder; MM-NCE performs better. Making the encoder trainable with MM-NCE yields further gains (+1.5% vs. frozen MM-NCE on STAR; +3.0% vs. matching+MSE), indicating that MM-NCE both aligns sets and adapts the relation encoder to video-specific patterns.

**Vision Queries vs. Hidden States**: Table 6.c compares tokens before vs. after the FFN; despite potential last-layer overfitting, the FFN-projected tokens perform best (STAR +7.4%, NExT-QA +1.2%, Intent-QA +1.8%), suggesting explicit relation tokens are more useful for LLMs.

| | STAR | NeXT-QA | Intent-QA |
|---|---|---|---|
| *a) Initialization:* | | | |
| Random init | 68.3 | 71.0 | 69.2 |
| RoBERTa-large | 68.0 | 72.2 | 71.0 |
| CLIP text encoder | 68.5 | 72.3 | 71.4 |
| Sentence embedder | **69.4** | **74.0** | **71.8** |
| *b) relations:* | | | |
| 1 | 65.3 | 72.4 | 70.5 |
| 2 | 68.3 | 72.9 | 70.7 |
| 4 | 68.0 | 73.1 | 71.3 |
| 8 | **69.4** | **74.0** | **71.8** |

Table 7: Ablation on a) the initialization of the relation encoder and b) the number of relations used as input to the LLM.

**Slow-Fast Architecture**: Table 6.d assess the impact of the dual-pathway architecture. It shows that using both representations consistently outperforms single-pathway variants (STAR: +0.5% over Slow, +4.2% over Fast), showing that modeling relations can be improved by detailed spatial information for object identification and efficient temporal modeling for action recognition.

**Relation Encoder Initialization**: Table 7.a demonstrates that initializing the relation encoder with a contrastively trained sentence embedder significantly outperforms alternatives (e.g., +0.9% over CLIP on STAR). This supports our claim that effective relation modeling requires semantically rich embeddings that can discriminate between similar but distinct relations (e.g., "person opens door" vs. "person closes door"), which contrastive training naturally provides.

**Number of Vision Queries**: Table 7.b shows that increasing from 1 to 8 relations per pathway consistently improves performance (STAR: +4.1%, NeXT-QA: +1.6% and Intent-QA: +1.3%), validating our modeling approach. This confirms that videos are better represented as sets of multiple relations rather than single global entities, with each additional relation adding new information.

| | STAR | NExT-QA |
|---|---|---|
| *(a) Top-k relation accuracy* | | |
| Top-1 | 53.7 | 30.0 |
| Top-3 | 83.7 | 58.5 |
| Top-5 | 90.6 | 72.5 |

Table 8: (Top-$k$ relation retrieval.

### 4.5 RELATION ACCURACY AND QUERY USAGE

**Relation retrieval accuracy.** To quantitatively assess the quality of the learned query–relation alignment, we evaluate REVEAL on STAR and NExT-QA as a relation retrieval task. For each video, we compute its vision queries, score them against the global pool of textual relation triplets (subject–predicate–object) using cosine similarity, and measure Top-$k$ accuracy at the video level: a video is counted as correct if at least one of its ground-truth relations appears among the Top-$k$ retrieved relations for at least one query. We observe, in table 8, high Top-$k$ accuracies, indicating that the learned queries reliably retrieve annotated relations.

**Vision query usage and structured video representations.** Finally, we analyze how different queries contribute to relation retrieval by reporting, for each query index, the percentage of relations in which that query yields a Top-5 hit. Results are presented in figure 4 as barplots. We find that multiple queries are used across videos, rather than a single dominant slot, which supports our claim that MM-NCE learns a compositional representation where distinct queries specialize to different relational aspects of the video.

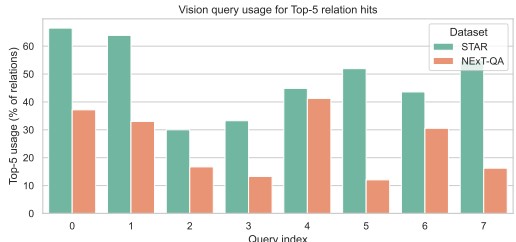

Figure 4: % of relations per query for which the query yields a top-5 hit.

## 5 CONCLUSION

We presented REVEAL, a framework advancing video understanding through relation-based representation learning. By modeling videos as relation triplet sets and introducing MM-NCE loss for aligning unordered relations, our approach creates structured embeddings that connect effectively with LLMs. Experiments show that relation-based representations outperform global ones.

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

## ACRONYMS

**BLIP** Bootstrapping Language-Image Pre-training

**CLIP** Contrastive Language-Image Pre-training

**CLS** Classification

**FFN** Feed Forward Network

**LLM** Large Language Model

**MIL-NCE** Multiple Instance Learning and Noise Contrastive Estimation

**MM-NCE** Many-to-Many Noise Contrastive Estimation

**MSE** Mean Squared Error

**REVEAL** RElation-based Video rEpresentAtion Learning

**VideoQA** Video-Question-Answering

**VLM** Video Language Models

## A  PRETRAINING DETAILS

### A.1  DATALOADING

Our data preprocessing and loading pipeline relies on WebDataset Aizman & Ott (2021). Pre-computed slow-fast CLIP features are stored as TAR files containing PyTorch Paszke et al. (2019) tensors. We use WebDataset's built-in shuffling mechanism with a buffer size of 5000 samples and an initial buffer of 1000 samples to ensure proper randomization.

### A.2  MODEL IMPLEMENTATION

The pretraining model architecture consists of dual-pathway transformers processing slow and fast video features. We extract CLIP's patch features from the penultimate layer for the slow pathway. Each pathway includes a projection layer that maps 1024-dimensional input features to a hidden dimension 768, followed by learnable positional encodings. The fast pathway processes CLS tokens features from 16 frames, while the slow pathway handles patch features from 4 frames. Both pathways utilize identical but separate transformer encoders, each comprising two encoder layers with 8-head self-attention (hidden size 768, FFN dimension $4 \times 768$). The model employs fixed positional encodings using sinusoidal functions. We implement separate embedding modules for relationship modeling, generating 8 learnable query embeddings for each pathway. The decoder architecture comprises 12 transformer decoder layers per pathway, each with 8-head cross-attention mechanisms and GELU activation functions. The decoder outputs are processed through an MLP with architecture $768 \to 4 \times 768 \to 1024$, where 1024 is the ground truth embedding dimension. The implementation includes careful initialization strategies: orthogonal initialization for query embeddings, normal initialization (mean=0, std=0.02) for linear layers, and zero for biases. All normalization layers use LayerNorm.

### A.3  PRETRAINING

Our pretraining implementation utilizes distributed training using PyTorch's DistributedDataParallel (DDP). The learning rate follows a cosine schedule with a linear warmup, starting from an initial learning rate of $lr = 0.00005$ with a 20% warmup period over total steps, decaying to $0.05 \times lr$ at completion. Training proceeds for 5 epochs with gradient accumulation every 4 steps and gradient clipping at 1.0. We implement a bidirectional contrastive loss adapted to our multi-prediction setting following open-clip implementation Cherti et al. (2023). We use the Hungarian matching implementation from Scipy Virtanen et al. (2020) to match predictions with ground truth. The model employs two separate prediction heads for slow and fast pathways, each producing embeddings of dimension 1024. We initialize the logit scale as $\log(1/0.07)$. We use the AdamW optimizer with a weight decay of 0.1. Training metrics are logged using Neptune.ai **?**, including gradient norms, learning rates, and various losses.

## B  Finetuning Details

Our implementation leverages Llama-VQA implementation Ko et al. (2023). Llama3 8B is the base language model, enhanced with REVEAL for video processing. We fine-tune using adapter layers while keeping the base Llama model frozen. Specifically, we use 32 adapter layers, with a length of tokens corresponding to the number of video relationships input to the LLM. The model extracts 16 relation queries per temporal segment, which are then linearly projected to match Llama's hidden dimension (4096). The training process uses AdamW optimizer with a base learning rate scaled by batch size (effective $lr = base\_lr \times batch\_size/256$), with a linear warmup over 2 epochs and cosine decay. The training is done for five epochs. We use slow-fast features with a dimension of 1024 for video features, which are processed through REVEAL before being integrated with the language model. For datasets requiring subtitles (TVQA and VLEP), we integrate them into the input sequence before the question. All video frame features are pre-extracted and stored. In table 9, we provide the hyperparameters per dataset.

| Hyperparameter | STAR | NextQA | Intent-QA | TVQA | VLEP |
|---|---|---|---|---|---|
| Base Learning Rate | 0.06 | 0.06 | 0.08 | 0.07 | 0.07 |
| Batch Size | 4 | 8 | 4 | 1 | 4 |
| Weight Decay | 0.14 | 0.1 | 0.14 | 0.02 | 0.12 |
| Temporal Resolution | 8 | 2 | 2 | 1 | 1 |
| Gradient Accum. | 8 | 4 | 4 | 4 | 2 |
| Bias | 3 | 3 | 3.5 | 3 | 3 |
| QAV loss | ✓ | ✓ | ✓ | ✓ | ✓ |
| VAQ loss | ✓ | ✓ | ✓ | ✓ | × |
| Max Sequence Length | 256 | 192 | 256 | 714 | 384 |

Table 9: Dataset-specific hyperparameters used in our experiments. Values were determined through empirical validation.

## C  Full Ablation Tables

Table 11 provides full per-category results for the temporal resolution, the relationship encoder initialization, and the number of relationships input to the LLM. The optimal temporal resolution, as expected intuitively, depends on the dataset. We also observe that the model with a relationship encoder initialized from a sentence embedder improves the performance of every question category evaluated. Finally, the more relationship vectors we input to the LLM, the better the results are, even though we get competitive results from a single relationship vector per temporal segment.

## D  Qualitative Analysis of the Performance on VideoQA

### D.1  Successful Cases

We present two successful examples from the STAR dataset where our model correctly answers the questions (Figure 5). In both cases, we visualize the alignment between the extracted relationship triplets and video segments (*i.e.*, the maximum similarity scores between the decoded queries and the encoded relationships) to demonstrate how REVEAL processes temporal information. In the first example, given the question "What is the person doing while eating a sandwich?", we extract the relationship triplet "Subject: person, Predicate: eating, Object: sandwich". The video is divided into 8 equal segments, and we observe strong alignment between this relationship and all segments, confirming the continuous eating action. The correct answer, "took blanket", shows increased alignment specifically during the relevant temporal window, while alternative choices exhibit lower alignment scores as these actions are absent in the video. In the second example, for the question "What happened before the person opened the door?", we observe that the question's relationship becomes well-aligned with the video during the final two segments, corresponding to the door-opening action. The correct answer, "sat at the table", shows stronger alignment during the first six segments, maintaining higher scores than incorrect choices.

### D.2 FAILURE CASES

We also analyze two failure cases from the STAR dataset (Figure 6) to understand the model's limitations. The first case involves question ambiguity: given "What is the person doing after touching the box?", the model predicts "put down the box" while the ground truth is "closed the box". The alignment plots show that both relationships are well-matched with the video content, suggesting that both answers could be valid interpretations of the observed action sequence. The second example ("What is the person doing after opening the closet?") demonstrates an object recognition challenge. While the correct answer involves taking a box, the model incorrectly predicts "take clothes". The alignment reveals the model's difficulty in recognizing the box, and the prediction may be influenced by Llama3's knowledge about items typically retrieved from closets.

## E PROMPT ENGINEERING AND RELATIONSHIP EXTRACTION

### E.1 PROMPT TEMPLATE

Figure 8 shows the complete prompt template used with Mistral-7B for relationship extraction. We leverage in-context learning by providing multiple examples of caption-relationship pairs before requesting the model to extract relationships from new captions. Each example demonstrates how to decompose a caption into subject-predicate-object triplets. The prompt includes diverse examples covering different types of actions, objects, and temporal relationships to encourage comprehensive extraction. This approach helps the model understand the expected format and granularity of the extracted relationships.

### E.2 RELATIONSHIP EXTRACTION EXAMPLES

Table 10 presents examples of relationships extracted from Webvid-2M captions, with corresponding video frames shown in Figure 7. The extraction results demonstrate several key properties of our approach. The LLM generates a focused set of core relationships for concise captions. In contrast, complex or longer captions yield more detailed relationship sets. The extracted relationships, while accurate, are not exhaustive - they do not cover every possible relationship that could be inferred from the video content. This non-exhaustive nature of the extracted relationships validates our design choice not to penalize missing relationships during training to let the model freely infer relevant relationship vectors from videos.

### E.3 RELATIONSHIP EXTRACTION PIPELINE FOR CHARADES AND VIDOR

The VidOR and Charades datasets provide temporal annotations of relationships between objects in videos. Each relationship is annotated by a subject-predicate-object triplet and its temporal extent (start and end frames). To process these relationships into meaningful clips, we first collect all temporal ranges $(t_{start}, t_{end})$ for each video. We then employ a dynamic grouping algorithm that identifies natural breaks in the temporal annotations by analyzing the gaps between consecutive relationships. Specifically, we calculate the gap sizes between temporally adjacent relationships and use the 75th percentile of these gaps as a threshold to determine significant temporal breaks. This approach naturally segments the video into clips containing temporally coherent relationships. For each resulting clip, we aggregate all relationships whose temporal extent overlaps with the clip's timeframe, creating a set of relationships that describe the scene dynamics within that temporal window.

## F LLM USAGE

LLM has been used to help with Latex table syntax, bibliography cleaning and to detect typos.

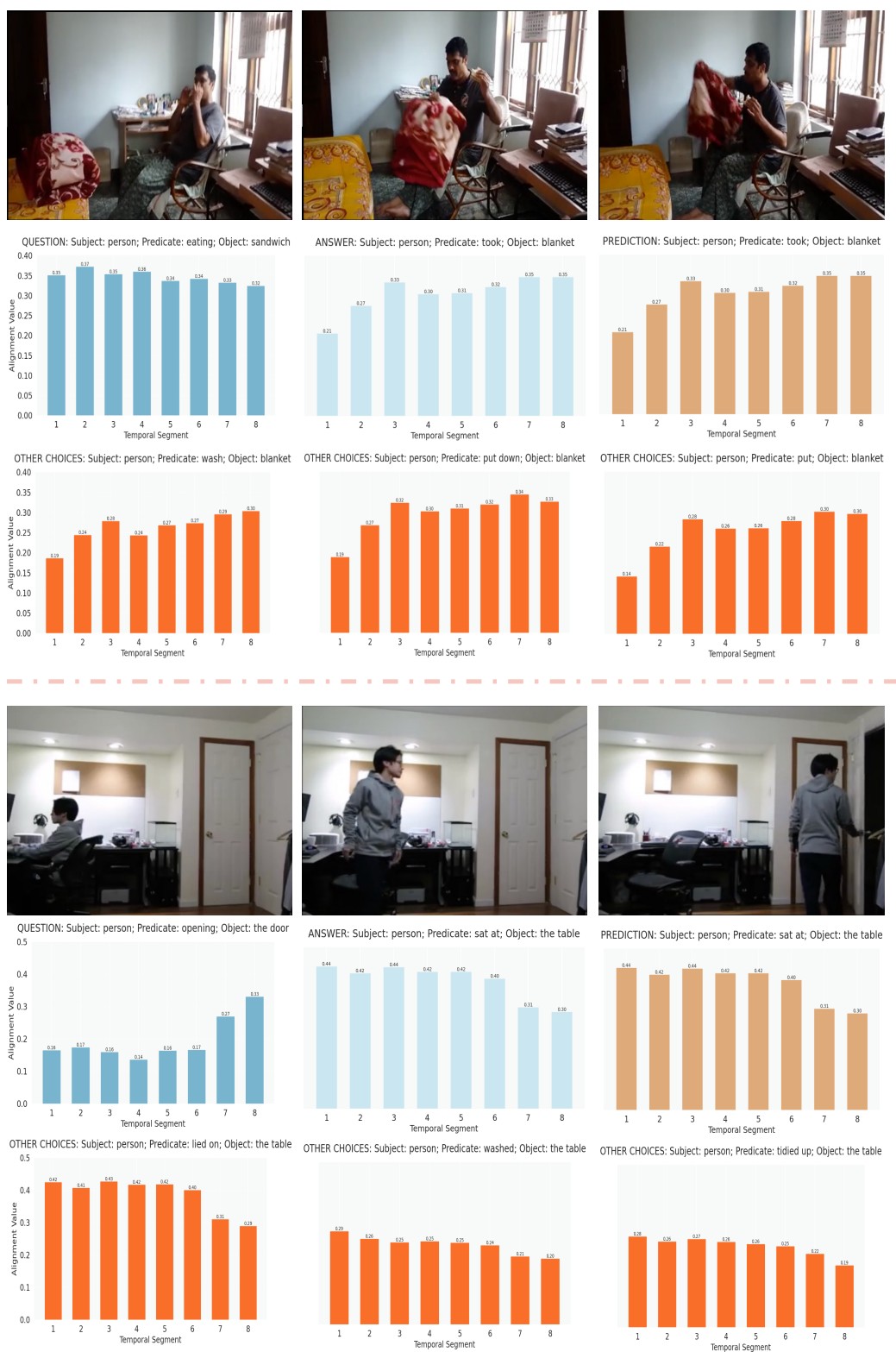

Figure 5: Successful examples from STAR dataset demonstrating REVEAL's relationship alignment capabilities. Top: The model correctly identifies concurrent actions (eating sandwich while taking blanket). Bottom: The model successfully captures temporal ordering of actions (sitting at table before opening door). Alignment scores between extracted relationships and video segments are visualized, showing stronger alignment during relevant temporal windows.

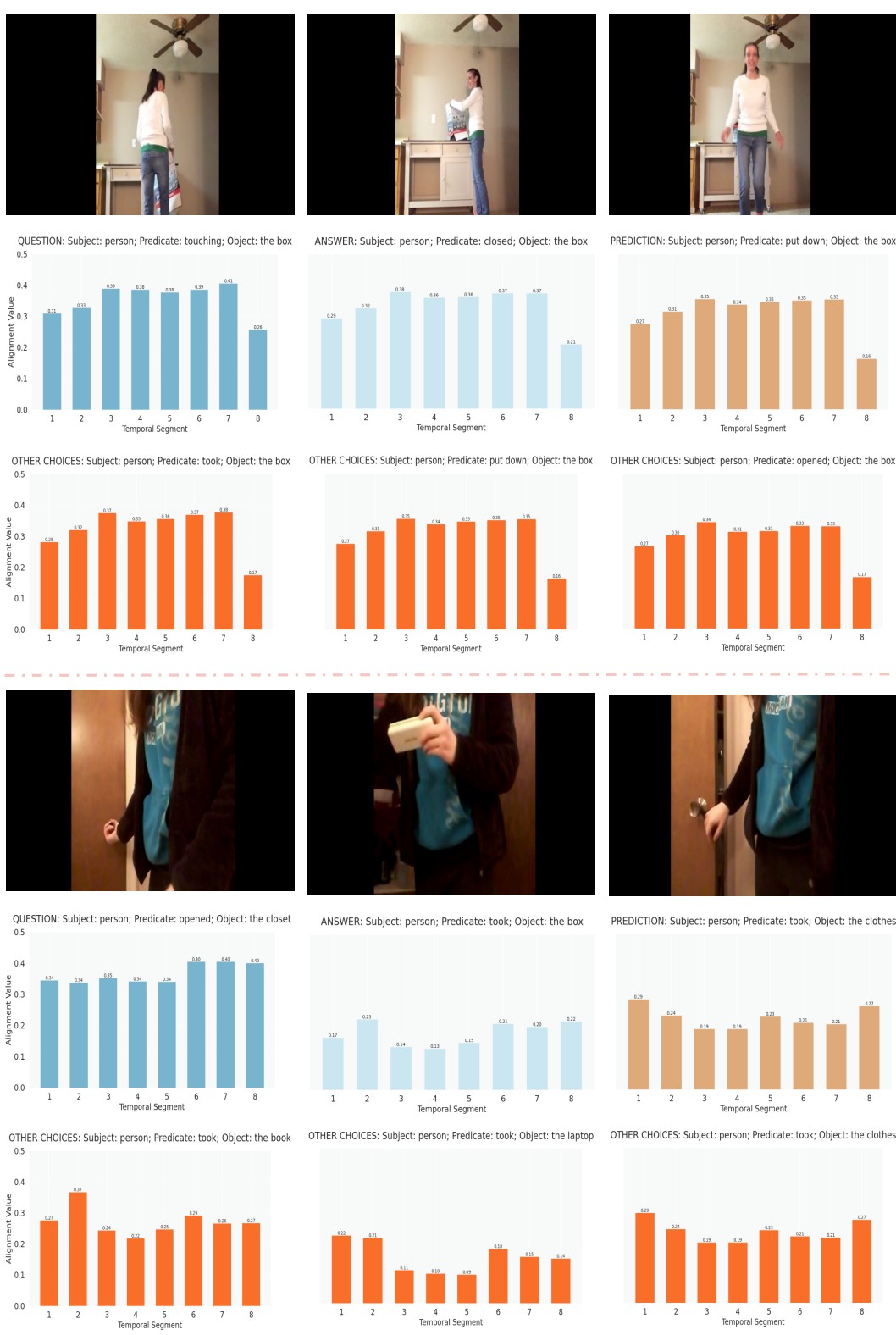

Figure 6: Failure cases from STAR dataset highlighting REVEAL's limitations. Top: Question ambiguity leads to multiple valid interpretations of the same action sequence. Bottom: Object recognition challenge where the model defaults to common-sense assumptions about closet contents rather than recognizing the specific object (small box).

| Caption | Relationships |
|---|---|
| Roses in blossom slow motion cinematic video | • Subject: roses , Predicate: in blossom, Object: none

• Subject: roses , Predicate: appearing in, Object: cinematic video
• Subject: cinematic video, Predicate: having, Object: slow motion |
| Male showing yellow particles inside the body
showing the cardiovascular system, lungs, heart,
liver, stomach and intestines with radar graphic
below and shining light from the top left corner | • Subject: male, Predicate: showing, Object: yellow particles

• Subject: male, Predicate: showing, Object: body

• Subject: body, Predicate: showing, Object: cardiovascular system
• Subject: body, Predicate: showing, Object: lungs

• Subject: body, Predicate: showing, Object: heart
• Subject: body, Predicate: showing, Object: liver
• Subject: body, Predicate: showing, Object: stomach
• Subject: body, Predicate: showing, Object: intestines
• Subject: radar graphic, Predicate: below, Object: male |
| Iguana on a tree hd | • Subject: iguana, Predicate: on, Object: tree |
| Turtle and tortoise on stone decoration design in
pond of garden japanese style in naritasan plum
garden of narita public park at chiba prefecture
in tokyo, japan | • Subject: turtle and tortoise, Predicate: on, Object: stone decoration
• Subject: turtle and tortoise, Predicate: in, Object: pond

• Subject: pond, Predicate: of, Object: garden

• Subject: garden, Predicate: japanese style, Object: None
• Subject: garden, Predicate: in, Object: Narita public park
• Subject: Narita public park, Predicate: at, Object: Chiba prefecture |
| Polishing of wooden plank using a rasp | • Subject: person, Predicate: polishing, Object: wooden plank |
| Athletic woman in sportswear holding feet on box
and doing evaluated reverse plank with leg raise
while training at outdoor fitness court | • Subject: athletic woman, Predicate: holding, Object: feet

• Subject: athletic woman, Predicate: doing, Object: reverse plank
• Subject: athletic woman, Predicate: raising, Object: leg
• Subject: box, Predicate: under, Object: feet
• Subject: outdoor fitness court, Predicate: at, Object: training |
| Canada goose family walking with the amazing
view of mount cook (aoraki) | • Subject: Canada goose family, Predicate: walking

• Subject: Canada goose family, Predicate: with, Object: amazing view
• Subject: amazing view, Predicate: of, Object: mount cook (aoraki) |
| Extreme close up image with chess game pieces
moved on the board by player hand | • Subject: player, Predicate: moving, Object: chess game pieces
• Subject: player, Predicate: taking, Object: chess game pieces
• Subject: chess game pieces, Predicate: on, Object: board
• Subject: image, Predicate: close up
• Subject: image, Predicate: containing, Object: chess game pieces and player hand
• Subject: image, Predicate: having, Object: extreme close up perspective |
| Aerial view of a beautiful beach with turquoise
water and waves crashing on the shore | • Subject: view, Predicate: aerial, Object: beach

• Subject: beach, Predicate: is, Object: beautiful
• Subject: water, Predicate: is, Object: turquoise
• Subject: waves, Predicate: crashing on, Object: shore |

Table 10: Video Captions From Webvid-2M and Their Extracted Relationships

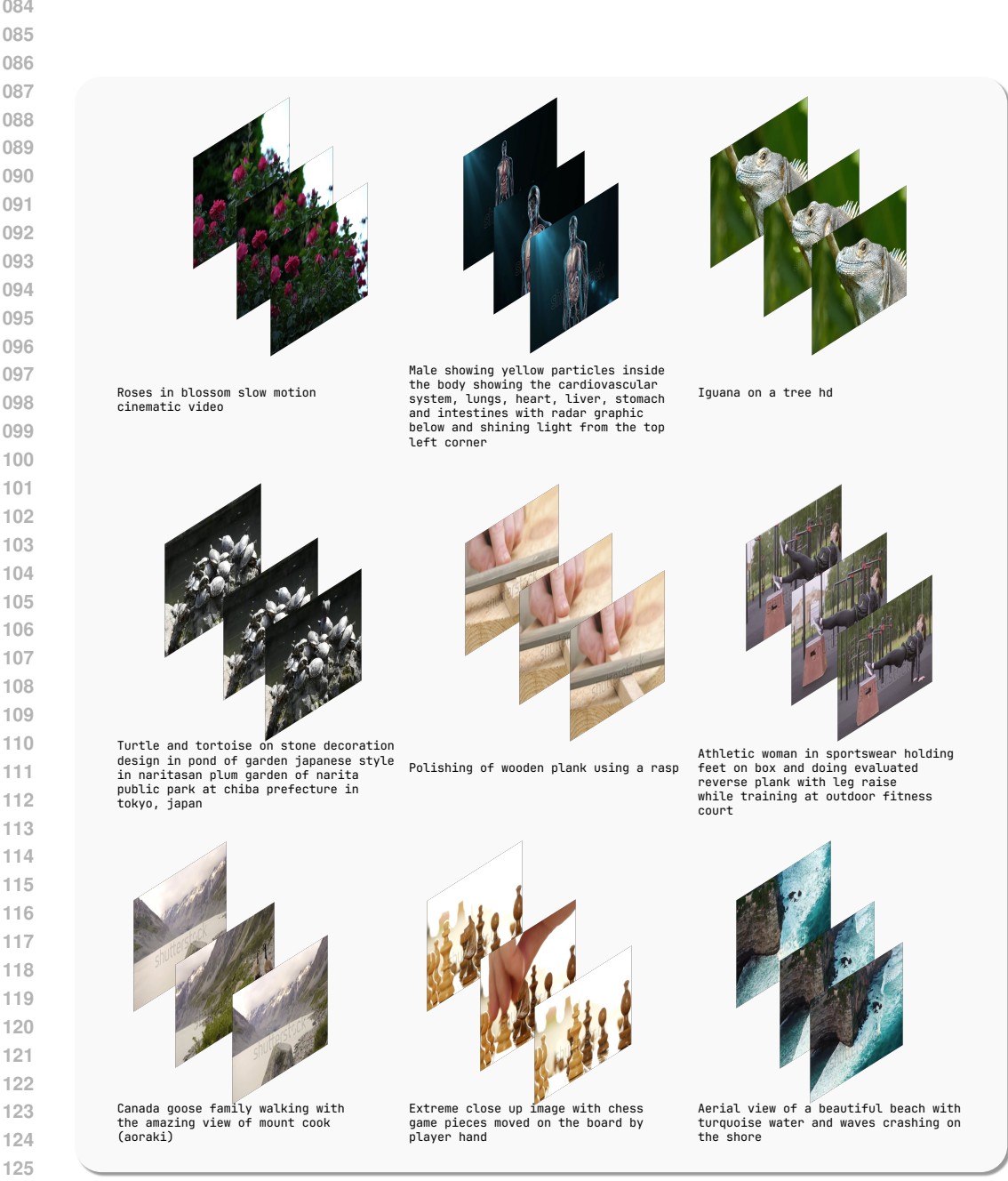

Figure 7: Sample videos from WebVid-2M

| Ablation | STAR | | | | | NeXT-QA | | | | Intent-QA | | | |
|---|---|---|---|---|---|---|---|---|---|---|---|---|---|
| | In | Seq | Pre | Feas | **All** | C | T | D | **All** | CW | CH | TP & TN | **All** |
| *a) Temp. Res.:* | | | | | | | | | | | | | |
| 1 | 54.9 | 62.3 | 64.1 | 65.9 | 61.8 | 74.2 | 68.8 | 77.0 | 73.4 | 74.3 | 61.0 | 55.0 | 70.7 |
| 2 | 57.6 | 65.1 | 69.4 | 68.4 | 65.1 | **74.0** | **70.0** | **77.9** | **73.3** | **74.6** | **75.5** | **65.6** | **71.8** |
| 4 | 59.2 | 68.0 | 70.7 | 69.0 | 66.7 | 73.5 | 69.2 | 76.7 | 72.6 | 74.3 | 74.7 | 60.3 | 71.1 |
| 8 | **61.4** | **69.3** | **75.0** | **72.0** | **69.4** | 73.7 | 69.6 | 75.4 | 72.7 | 72.2 | 75.0 | 60.6 | 70.8 |
| *b) Rel. Init.:* | | | | | | | | | | | | | |
| Random | 59.7 | 68.2 | 72.9 | 72.5 | 68.3 | 72.4 | 67.1 | 74.4 | 71.0 | 70.7 | 73.4 | 58.7 | 69.2 |
| RoBERTa-large | 60.4 | 67.9 | 74.0 | 69.6 | 68.0 | 73.8 | 68.4 | 74.9 | 72.2 | 73.7 | 75.9 | 60.2 | 71.0 |
| CLIP text encoder | 59.4 | 69.0 | 73.1 | 72.5 | 68.5 | 74.0 | 68.2 | 74.9 | 72.3 | 72.2 | 75.9 | 61.0 | 71.4 |
| Sentence embedder | **61.4** | **69.3** | **75.0** | **72.0** | **69.4** | 74.0 | 70.0 | 77.9 | 73.3 | 74.6 | 75.5 | 65.6 | 71.8 |
| *c) #Rels:* | | | | | | | | | | | | | |
| 1 | 57.8 | 66.9 | 68.4 | 67.8 | 65.3 | 73.8 | 68.0 | 77.2 | 72.4 | 72.5 | 74.1 | 60.8 | 70.5 |
| 2 | 62.1 | 68.4 | 74.0 | 68.4 | 68.3 | 74.1 | 68.9 | 77.1 | 72.9 | 74.9 | 74.3 | 60.2 | 70.7 |
| 4 | 60.4 | 67.7 | 73.7 | 70.2 | 68.0 | **75.1** | 68.3 | 75.9 | 73.1 | **74.9** | 74.4 | 61.8 | 71.3 |
| 8 | **61.4** | **69.3** | **75.0** | **72.0** | **69.4** | 74.0 | **70.0** | **77.9** | **73.3** | 74.6 | **75.5** | **65.6** | **71.8** |

Table 11: Full per category results for a) Temporal resolution; b) Relationship encoder initialization and; c) number of relationships vectors input to the LLM.

---

**Relationships Extraction Prompt**

```
[INST] You are a software to extract relationships from sentences.
Extract explicit and factual relationships between objects in the last sentence.
Use the same formatting as below. No other text.
One instance per subject, object, and predicate. Be exhaustive.

Sentence: 'A video of a person on the  side of a table holding food.'
subject: person, predicate: on the side of, object: table
subject: person, predicate: holding, object: food

Sentence: 'A kid touching the table  while sitting on a chair.'
subject: kid, predicate: touching, object: table
subject: kid, predicate: sitting on, object: chair

Sentence: 'A man putting on shoes and clothes.
Behind him two trees next to each other.'
subject: man, predicate: holding, object: shoe
subject: man, predicate: holding, object: clothes
subject: two trees, predicate: behind, object: him
subject: tree, predicate: next to, object: tree

Sentence: 'Woman sets table with plates, silverware, glasses,
before placing oatmeal pot and juice pitcher in center. Calls family.'
subject: woman, predicate: set, object: table
subject: woman, predicate: set, object: plates
subject: woman, predicate: set, object: silverware
subject: woman, predicate: set, object: glasses
subject: woman, predicate: placing, object: oatmeal pot
subject: woman, predicate: placing, object: juice pitcher
subject: oatmeal pot, predicate: in center of, object: table
subject: juice pitcher, predicate: in center of, object: table
subject: woman, predicate: call, object: family

Sentence: 'Children playing on swings and slide. Couple sits on bench,
holding hands.'
subject: children, predicate: playing on, object: swings
subject: couple, predicate: sit on, object: bench
subject: couple, predicate: holding, object: hands [/INST]
Sentence: {sentence}
```

Figure 8: Prompt for Extracting Relationships from Sentences

