# OpenReview forum: "REVEAL: Advancing Relation-based Video Understanding for Video-Question-Answering"
_ICLR.cc/2026/Conference — Submitted to ICLR 2026_

### Official Review · Reviewer_UJzk · 2025-10-20

**Soundness:** 2
**Presentation:** 2
**Contribution:** 2
**Rating:** 2
**Confidence:** 5

**Summary:**

The paper presents REVEAL (RElation-based Video rEpresentAtion Learning), a framework for Video Question Answering (VideoQA) that models videos as temporal relation triplets (subject–predicate–object) derived from captions. A Q-Former encodes video frames into query embeddings aligned with textual relations through a new Many-to-Many Noise Contrastive Estimation (MM-NCE) loss, enabling fine-grained visual–text alignment. Experiments on five benchmarks (NeXT-QA, Intent-QA, STAR, VLEP, TVQA) demonstrate competitive performance. Key contributions include relation-based video encoding, the MM-NCE alignment loss, and extensive evaluation across multiple datasets.

**Strengths:**

The paper is original in reformulating VideoQA through relation-based video representation and introducing the MM-NCE loss for aligning unordered multimodal sets. It creatively integrates ideas from scene graphs and contrastive learning into a unified, scalable framework.

**Weaknesses:**

The method’s novelty is limited. Theoretical justification for MM-NCE and evidence of its alignment correctness are lacking. Experimental gains are modest, with missing comparisons to recent methods. Writing clarity, figure details, and reference formatting also need significant improvement.

**Questions:**

1. The paper has significant issues in terms of writing format (e.g., reference citation style) as well as overall logical flow and clarity of presentation. For instance, the textual details in Figures 1 and 2 are unclear and require careful revision. Substantial effort is needed to improve the overall readability and structure.

2. The main claimed contribution lies in introducing an additional pretraining stage to enhance performance on the VideoQA task. Regardless of the level of novelty, the proposed pretraining essentially serves as a form of fine-grained alignment. Has the paper compared this approach with other fine-grained alignment methods? Moreover, in the Many-to-Many Noise Contrastive Estimation module, how is the alignment correctness between q and r ensured? Is there any theoretical or qualitative analysis to support this design?

3. From the experimental results, the proposed method does not demonstrate clear advantages over prior work—particularly in Table 2. In addition, several tables lack comparisons with the latest state-of-the-art methods.

---

> ### Author Response · Authors · 2025-11-25
> **Response to Reviewer UJzk**
>
> We thank the reviewer for the detailed feedback. Below, we address the concerns on novelty, MM-NCE justification, empirical advantages, and presentation.
>
> ---
>
> ### W1. Novelty: beyond “just an additional pretraining stage”
>
> > “The main claimed contribution lies in introducing an additional pretraining stage… Has the paper compared this approach with other fine-grained alignment methods?”
>
> Our main contribution is not only the existence of a pretraining stage, but the objective and representation it learns:
>
> - **Set-level, open-vocabulary relation alignment:**
>
>     REVEAL models each video as an **unordered, variable-length set** of relation triplets and aligns these to a fixed set of vision queries. The proposed **MM-NCE loss**:
>
>     - uses **Hungarian matching** to find an injective assignment from text relations to visual queries for each video, and
>     - applies a **symmetric contrastive objective** in both query→relation and relation→query directions.
>     This is designed specifically for **unordered, non-exhaustive**, open-vocabulary relations, which differ from standard caption-based NCE or simple token-wise alignment.
> - **Comparison to standard (fine-grained) global alignment:**
>
>     In Table 6.a, we directly compare:
>
>     - **Captions + NCE loss**, *i.e.*, standard caption-based alignment, with attention pooling over queries, against
>     - **Relations + MM-NCE loss**, *i.e.*, our relation-based objective.
>
>     Under strictly identical data/backbone, relation-based MM-NCE yields large gains, demonstrating that going from global caption alignment to **structured relation supervision with our loss** is a substantive methodological change, not just extra pretraining. These results suggest that video-relation modeling is not necessarily more data-efficient than video-global caption modeling. We note that pretraining occurred only on ~3 million samples, pointing out the significant potential of the proposed method in a larger-scale setting.
>
>
> | Pretraining Setup | STAR All | NExT-QA All | Intent-QA All |
> | --- | --- | --- | --- |
> | Captions + NCE loss | 31.5 | 56.3 | 61.2 |
> | Relations + MM-NCE loss | 65.4 | 72.8 | 70.8 |
> - **Orthogonal to architecture:**
> We plug REVEAL into existing VideoQA pipelines (Flipped-VQA, Vamos) with **identical vision and language backbones**, keeping the video encoder frozen during fine-tuning. This isolates the contribution of the relation-based objective and representation, as shown across five benchmarks.

---

> > ### Author Response · Authors · 2025-11-25
> > **Response to Reviewer UJzk (Part 2)**
> >
> > ### Q2. MM-NCE justification and alignment “correctness”
> >
> > > “How is the alignment correctness between q and r ensured? Is there any theoretical or qualitative analysis to support this design?”
> >
> > Our design combines **per-video injective relation-matching** with **noise contrastive estimation learning**.
> >
> > - **Mechanism:**
> >     - For each video $k$, Hungarian matching computes an injective mapping $\sigma^{(k)}$ that maximizes the sum of cosine similarities between relation embeddings and vision queries, over the set $\mathcal{S}_{J^{(k)},M}$ of injective maps.
> >     - MM-NCE then defines two symmetric InfoNCE terms:
> >         - $L_{q \rightarrow r}$, which pushes each matched query closer to its assigned relation and away from all relations in the batch, and
> >         - $L_{r \rightarrow q}$, which pushes each relation closer to its assigned query and away from all queries in the batch.
> >     - Only **matched pairs** contribute to the positive term; unmatched queries are **not penalized** for not matching any current relation, which allows for non-exhaustive supervision.
> > - **Empirical evidence vs. alternative objectives (Table 6.b):**
> >     - We compare:
> >         - **Frozen + MSE loss** (Hungarian matching + regression),
> >         - **Frozen + MM-NCE**, and
> >         - **Trainable + MM-NCE**.
> >         MM-NCE consistently outperforms MSE after matching, and further improving the relation encoder's trainability further improves performance on all datasets. This indicates that:
> >         1. The contrastive formulation is more effective than simple regression on matched pairs, and
> >         2. The encoder and queries **learn** the alignment rather than merely reflecting a fixed prior.
> >
> > | Relation Encoder + Loss | STAR All | NExT-QA All | Intent-QA All |
> > | --- | --- | --- | --- |
> > | Frozen + MSE loss | 66.4 | 71.1 | 68.9 |
> > | Frozen + MM-NCE loss | 67.9 | 72.6 | 71.4 |
> > | Trainable + MM-NCE loss | 69.4 | 74.0 | 71.8 |
> > - **Alignment with incomplete labels:**
> > Because the mapping is injective and non-surjective when $J^{(k)} < M$, some queries remain unmatched and do not receive positive gradients. These queries are free to represent additional video content (potentially unannotated relations) while still acting as **negatives** for other videos’ relations in the batch. Table 7 b shows that each vision-query contains original information about the video it encodes: we get the best results when we provide the full set of queries to the LLM. This demonstrates the compositional nature of the learned video-representation that directly stems from MM-NCE.
> >
> > | # Relations to LLM | STAR All | NExT-QA All | Intent-QA All |
> > | --- | --- | --- | --- |
> > | 1 | 65.3 | 72.4 | 70.5 |
> > | 2 | 68.3 | 72.9 | 70.7 |
> > | 4 | 68.0 | 73.1 | 71.3 |
> > | 8 | 69.4 | 74.0 | 71.8 |
> >
> > To complement these ablations, we now report a direct relation-retrieval evaluation on STAR and NExT-QA. For each video clip, we first construct a global label space of textual relation triplets (subject–predicate–object) from all STAR annotations and encode each relation once with the REVEAL model’s text encoder. For a given video, we compute its REVEAL vision queries and compute cosine similarities between every query and every relation class. For each query, we rank all relation classes by similarity and define Top‑k accuracy at the video level as the fraction of videos where at least one ground‑truth relation appears in the Top‑k predictions of at least one query. As shown in the following tables, REVEAL performs strongly on these tasks, demonstrating the quality of the video-relation pretraining. Furthermore, we report the usage percentage of vision queries during relation retrieval (i.e., the proportion of times a given query is used to retrieve the ground-truth relation). As we can see, all queries are used, and query usage is rather balanced, highlighting the structured and composed properties of these representations derived from video-relation modeling.
> >
> > - accuracy
> >
> > | k | STAR | Next‑QA |
> > | --- | --- | --- |
> > | 1 | 53.69 | 30.02 |
> > | 3 | 83.69 | 58.53 |
> > | 5 | 90.58 | 72.51 |
> >
> > - Query usage
> >
> > | Query Index | STAR | Next‑QA |
> > | --- | --- | --- |
> > | 0 | 66.48% | 37.16% |
> > | 1 | 63.87% | 33.00% |
> > | 2 | 30.00% | 16.69% |
> > | 3 | 33.25% | 13.27% |
> > | 4 | 44.86% | 41.26% |
> > | 5 | 51.93% | 12.04% |
> > | 6 | 43.58% | 30.54% |
> > | 7 | 54.79% | 16.23% |

---

> ### Author Response · Authors · 2025-11-25
> **Response to Reviewer UJzk (Part 3)**
>
> ### Q3. Empirical strength and SOTA coverage
>
> > “From the experimental results, the proposed method does not demonstrate clear advantages over prior work—particularly in Table 2. In addition, several tables lack comparisons with the latest state-of-the-art methods.”
> >
>
> We agree that fairness and clarity in comparisons are important, and our tables are organized to ensure a completely fair apples-to-apples comparison. Our experimental protocol is to "plug" our method into existing settings to fairly compare our approach to baselines using the same ViT backbone and LLM.
>
> - **STAR (Table 1):** With Llama1/ViT-L/14 and the same fine-tuning protocol, REVEAL improves over Flipped-VQA by **+2.5%** overall, and even achieves stronger results than the larger ViT-G/14 backbone-based method.
> - **NExT-QA (Table 2):** With Llama1/ViT-L/14, REVEAL reaches **72.7%** vs. 72.0% for Flipped-VQA. With Llama3, REVEAL further improves to **74.0%**.
> - **Intent-QA (Table 3):** REVEAL attains **72.8%**, surpassing Flipped-VQA (69.5%) by **+3.3%** with identical backbones.
> - **TVQA and VLEP (Tables 4, 5):** REVEAL improves over Flipped-VQA by **+0.8%** (83.0 vs. 82.2) on TVQA and by **+1.2%** (73.5 vs. 72.3) on VLEP with Llama3/ViT-L/14.
> - **Comparison to Vamos with captions:**
>
>     For a fair comparison to methods that leverage captions more heavily, we add a **REVEAL+Captioning** variant. On NExT-QA, this reaches **77.2%**, on par with Vamos (77.3%), and on Intent-QA, REVEAL+Captioning reaches **75.0%**, surpassing Vamos (74.1%) with the same backbones.
>
>
> In Table 2, we include results for recent VLMs trained with larger backbones (e.g., SigLIP, CLIP ViT-G) and large-scale instruction-tuning data, sometimes including data closely related to the evaluation benchmarks. We explicitly label their specifications (language backbone, vision backbone, IT/ZS vs PT/FT regimes) and group them per these criteria. These results are thus provided as references rather than direct baselines.
>
> ---
>
> ### W2 and Q1. Presentation and clarity
>
> > “The paper has significant issues in terms of writing format (e.g., reference citation style) as well as overall logical flow and clarity of presentation. For instance, the textual details in Figures 1 and 2 are unclear… Substantial effort is needed to improve the overall readability and structure.”
>
> We appreciate this feedback and are committed to improving the presentation along several key axes. We already updated figures, for better visibility, and fixed citations in the revised version. We commit to making further improvements to writing quality and the method introduction for the camera-ready version, as well as addressing suggestions and requests.
>
> We hope these clarifications and planned improvements address your concerns regarding novelty, justification for alignment, empirical strength, and presentation.

---

> > ### Comment · Reviewer_UJzk · 2025-11-26
> >
> > I acknowledge the authors' efforts in addressing some of the reviewers' comments within the constrained time. However, the newly added experiments, while helpful, do not fundamentally alter my assessment regarding the limited methodological novelty of this work. Having considered the authors' rebuttal and the other reviews, I still do not find sufficient grounds to raise my score. I encourage the authors to thoroughly address the pertinent issues raised by all reviewers in a future submission.

---

### Official Review · Reviewer_NhbL · 2025-10-31

**Soundness:** 3
**Presentation:** 3
**Contribution:** 3
**Rating:** 6
**Confidence:** 3

**Summary:**

The paper "REVEAL: Advancing Relation-Based Video Understanding for Video-Question-Answering" introduces a novel framework that significantly enhances VideoQA by modeling video content as a structured collection of temporal (subject-predicate-object) relation triplets, departing from traditional global video representations. A central technical innovation is the Many-to-Many Noise Contrastive Estimation (MM-NCE) loss, which elegantly aligns unordered and incomplete sets of visual queries with text-derived relation descriptions using Hungarian matching, thereby robustly learning from non-exhaustive, web-supervised data. This modular framework, comprising dual-pathway vision encoders, temporal encoders, a Relation Q-Former, and a text-based Relation Encoder, seamlessly integrates with Large Language Models (LLMs) via adapters. Comprehensive experimental evaluation across five challenging VideoQA benchmarks (NeXT-QA, Intent-QA, STAR, VLEP, TVQA) demonstrates REVEAL's competitive or superior performance, especially in tasks requiring deep temporal and relational reasoning, underscoring the efficacy and promise of its relation-centric paradigm for more interpretable and robust video understanding.

**Strengths:**

1.REVEAL introduces a unique and promising paradigm by explicitly modeling video content as sets of temporal (subject-predicate-object) relation triplets. This contrasts with traditional global or patch-token representations, offering a finer-grained, more interpretable approach to capture interactions and evolution within videos, crucial for complex temporal and relational reasoning in VideoQA.
2.The proposed MM-NCE loss is a key technical innovation. It cleverly addresses the challenges of unordered set alignment and incomplete annotations in video-language learning. By using Hungarian matching for optimal query-relation correspondence in contrastive learning, MM-NCE efficiently learns from non-exhaustive, web-supervised relation data, enabling the model to infer relevant relationships and learn more general representations without penalizing missing annotations.
3.REVEAL demonstrates its effectiveness through extensive evaluation across five challenging VideoQA benchmarks (NeXT-QA, Intent-QA, STAR, VLEP, TVQA). It achieves competitive, and in many cases superior, performance against state-of-the-art models, particularly on tasks requiring temporal reasoning and relation comprehension. This robust experimental validation reinforces the benefits of relation-based representations for complex video semantics.
4.The REVEAL framework boasts a modular design, with distinct visual, temporal, Q-Former, and relation encoders. This architecture offers high flexibility and scalability, allowing visual features to be transformed into efficient tokens that seamlessly integrate with existing LLMs via adapters. The use of Mistral-7B for web-supervised relation extraction further highlights its potential for efficient knowledge distillation. This design facilitates future advancements and broader applications.

**Weaknesses:**

1.The core claim that relation triplets offer superior, structured representations for VideoQA, especially when connecting with LLMs, needs stronger evidence. While Table 6.a shows improved performance over caption-based NCE, it doesn't adequately demonstrate why (subject-predicate-object) triplets are intrinsically better or more efficient than other sophisticated global video representations or longer, richer textual descriptions.
2.The paper claims that its design choice to "not penalize missing relationships during training" is validated by the non-exhaustive nature of extracted relations. However, the theoretical underpinnings of how MM-NCE effectively handles incomplete relation sets without introducing noise or bias, and how the model reliably infers relevant relationships from unannotated content, require deeper theoretical analysis and empirical validation beyond a mere assertion.
3.While MM-NCE is presented as a solution for aligning unordered and incomplete sets, its unique theoretical advantages and practical distinctions from other advanced multi-to-multi matching or contrastive learning techniques (e.g., graph-matching, attention-based, or Transformer-based alignment) are not thoroughly analyzed. The mechanism for handling asymmetric set sizes (J(k) < M and M < J(k)) also needs clearer explanation.
4. The use of Mistral-7B for automated relation triplet extraction lacks comprehensive evaluation regarding its robustness, accuracy, and potential biases (e.g., preference for certain verbs, errors in complex sentences, or propagating dataset stereotypes). The quality gap between these automatically generated relations and human annotations, particularly with ambiguous video captions, is a significant unaddressed weakness.

**Questions:**

1.How does the explicit relation triplet representation intrinsically outperform other sophisticated global video representations for LLM-based VideoQA, considering potential information loss from unstructured captions?
2.How does MM-NCE's "free inference" for unannotated relations ensure robustness and prevent learning inaccurate or hallucinatory relationships, beyond merely not penalizing missing ones?
3.What are the precise theoretical and empirical distinctions of MM-NCE from other advanced multi-to-multi matching or contrastive learning techniques, especially regarding asymmetric set sizes and one-to-one correspondence?
4.Could you provide quantitative evaluation metrics (e.g., F1 score) for the Mistral-7B relation extraction against a reference set, and how does its quality impact downstream VideoQA performance and robustness?

---

> ### Author Response · Authors · 2025-11-25
> **Response to Reviewer NhbL**
>
> We thank the reviewer for the positive evaluation of our work. Especially the view of REVEAL’s relation-centric formulation as a promising paradigm for capturing temporal structure beyond traditional global and patch-token representations, and that you recognize MM-NCE as a key technical contribution for aligning unordered and non-exhaustive sets of visual queries and text relations. We also appreciate your emphasis on our extensive empirical evaluation.
>
> ---
>
> ### W1 and Q1. Why relation triplets vs. global/longer textual representations?
>
> > “How does the explicit relation triplet representation intrinsically outperform other sophisticated global video representations for LLM-based VideoQA, considering potential information loss from unstructured captions?”
> >
>
> Our goal is to expose LLMs to **structured, decomposed** video semantics rather than a single global vector or long caption, and we validate this both conceptually and empirically.
>
> - **Conceptually**, global video caption alignment is prone to “bag-of-words” behavior, where fine-grained interactions and role assignments (who does what to whom) are blurred. We explicitly discuss this in Sec. 3 and Sec. 4, referencing analyses such as [1] that show CLIP-style text encoders can behave like bags-of-words and underuse syntactic structure. By decomposing captions into (subject, predicate, object) triplets, we force the model to represent atomic relations (e.g., “person opens door”, “person closes door”) as separate supervision targets rather than compressing them into one vector.
> - Empirically, Table 6a (“Video-Relation vs. Video-Caption Alignment”) provides a direct comparison between:
>     - training on **captions + NCE** (standard global caption alignment), and
>     - training on **relations + MM-NCE** (our proposed set-based relation supervision),
>     under the same data and backbone:
>
>     | Supervision | STAR All | NExT-QA All | Intent-QA All |
>     | --- | --- | --- | --- |
>     | Captions + NCE loss | 31.5 | 56.3 | 61.2 |
>     | Relations + MM-NCE loss | 65.4 | 72.8 | 70.8 |
>
>     This shows large and consistent gains when moving from global captions to structured relation triplets.
>
> - **Complementarity with longer text**: We also evaluate a variant that combines relations with captions (REVEAL+C). On NExT-QA and Intent-QA, REVEAL+C matches or surpasses strong caption-based baselines such as Vamos using the same backbone (e.g., 77.2 vs. 77.3 on NExT-QA, 75.0 vs. 74.1 on Intent-QA). This demonstrates that REVEAL provides structured video representations that can be augmented with richer descriptions when needed.
>
> Together, these results support our claim that triplet-based supervision yields more effective and efficient video representations for LLM-based VideoQA than relying solely on global or long textual descriptions.
>
> [1] Yuksekgonul, Mert, et al. "When and why vision-language models behave like bags-of-words, and what to do about it?" In ICLR, 2023.

---

> ### Author Response · Authors · 2025-11-25
> **Response to Reviewer NhbL (Part 2)**
>
> ### W2 and Q2. Handling incomplete relation sets, “free inference”, and asymmetric sizes
>
> > *“How does MM-NCE’s ‘free inference’ for unannotated relations ensure robustness and prevent learning inaccurate or hallucinatory relationships, beyond merely not penalizing missing ones?”*
>
> - **Injective matching with non-exhaustive labels.**
>
>   For each video, we have a set of text-derived relation embeddings and a fixed set of vision queries. We compute the optimal **injective mappings** from relations to queries (Sec. 3.4). This creates a set of one-to-one **positive pairs** per video. Specifically, when $J^{(k)} < M$, **not all queries are matched**; unmatched queries are simply **omitted from the positive term** of the loss (Sec. 3.4, discussion on non-surjectivity and non-exhaustive annotation).
>
> - **Symmetric MM-NCE and robustness to missing labels.**
>
>   The MM-NCE loss consists of two symmetric InfoNCE terms, $L_{q \rightarrow r}$ and $L_{r \rightarrow q}$, summing over all matched pairs but using all other relations/queries in the batch as negatives. This has two key effects: unnmatched queries are not penalized for not corresponding to any current caption relation, which is important under non-exhaustive, web-derived triplets. At the same time, all queries participate as negatives for other videos’ relations. This prevents them from collapsing to arbitrary or hallucinatory semantics: they must remain distinct from relations that are clearly absent in their own video (since those appear as positives elsewhere in the batch). When we ablate on the number of vision queries provided to the LLM during VideoQA finetuning, performance consistently improve the more queries we use until we use all the vision queries. This proves that even in this assymetric and non-exhaustive setting, MM-NCE allows for robust and compositional video representations.
>
> | # Relations to LLM | STAR All | NExT-QA All | Intent-QA All |
> |--------------------|----------|-------------|---------------|
> | 1                  | 65.3     | 72.4        | 70.5          |
> | 2                  | 68.3     | 72.9        | 70.7          |
> | 4                  | 68.0     | 73.1        | 71.3          |
> | 8                  | 69.4     | 74.0        | 71.8          |
>
> - **Asymmetric sizes $J > M$.**
>
>   In practice, we fix the number of vision queries $M$ and, if more than $M$ relations are extracted, we randomly sample up to $M$ text relations (Sec. 4.3: “If more than eight text-derived relation embeddings are available, we randomly sample eight triplets.”). This ensures a well-defined injective mapping while still providing multiple, diverse relations per video over the course of training.
>
> We further evaluated the learned relation space as a query–relation retrieval task on STAR and NExT-QA. For each video, we compute REVEAL’s vision queries, score them against all relation classes, and measure whether at least one ground-truth relation appears in the Top‑k predictions of at least one query. We demonstrate strong relation-retrieval performance and we also report per-query usage statistics show that multiple queries are responsible for Top‑5 hits, rather than a single dominant slot. These results provide quantitative evidence that MM-NCE learns meaningful, compositional query–relation alignment: different queries specialize to different relations, and at least one query reliably captures ground-truth relations, supporting our claims about set-level alignment and “free inference” on unannotated content.
>
> - accuracy
>
> | k  | STAR | Next‑QA |
> |----|---------------------|------------------------|
> | 1  | 53.69              | 30.02                 |
> | 3  | 83.69              | 58.53                 |
> | 5  | 90.58              | 72.51                 |
>
> - Query usage
>
> | Query Index | STAR | Next‑QA |
> |-------------|-----------------------------------------|-------------------------------------------|
> | 0           | 66.48%                                  | 37.16%                                    |
> | 1           | 63.87%                                  | 33.00%                                    |
> | 2           | 30.00%                                  | 16.69%                                    |
> | 3           | 33.25%                                  | 13.27%                                    |
> | 4           | 44.86%                                  | 41.26%                                    |
> | 5           | 51.93%                                  | 12.04%                                    |
> | 6           | 43.58%                                  | 30.54%                                    |
> | 7           | 54.79%                                  | 16.23%                                    |

---

> > ### Author Response · Authors · 2025-11-25
> > **Response to Reviewer NhbL (Part 3)**
> >
> > ### W3 and Q3. Distinction from other multi-to-multi alignment methods
> >
> > > “What are the precise theoretical and empirical distinctions of MM-NCE from other advanced multi-to-multi matching or contrastive learning techniques, especially regarding asymmetric set sizes and one-to-one correspondence?”
> > >
> >
> > We explicitly design MM-NCE to differ from prior **multi-instance** or **global** alignment schemes by enforcing one-to-one query–relation correspondences.
> >
> > - **Compared to MIL-NCE.**
> > As discussed in Sec. 3.4, MIL-NCE is formulated to align **multiple captions to a single representation**, often via max-pooling or attention over instances, and does not enforce a one-to-one mapping between multiple video “slots” and multiple textual relations. In contrast:
> >     - Our Hungarian step finds a **discrete assignment** between individual relations and individual queries for each video.
> >     - The symmetric InfoNCE is applied **per matched pair**, ensuring each query is trained to specialize on a particular relation (slot behavior) rather than contributing indistinguishably to a single global embedding.
> >     We highlight this in the text: “Unlike MIL-NCE, designed to align multiple captions to a single representation, this approach enforces a one-to-one correspondence between the multiple video representations (the vision queries) and the corresponding text relations.”
> > - **Compared to simple matching + regression.**
> > Table 6b empirically compares **“Frozen + MSE loss”** (matching + regression) with **“Frozen + MM-NCE”** and **“Trainable + MM-NCE”**. The latter consistently outperforms MSE-only matching across all datasets, showing that:
> >
> > | Relation Encoder + Loss | STAR All | NExT-QA All | Intent-QA All |
> > | --- | --- | --- | --- |
> > | Frozen + MSE loss | 66.4 | 71.1 | 68.9 |
> > | Frozen + MM-NCE loss | 67.9 | 72.6 | 71.4 |
> > | Trainable + MM-NCE loss | 69.4 | 74.0 | 71.8 |
> >
> > ---
> >
> > ### W4 and Q4. Robustness, accuracy, and bias of Mistral-7B relation extraction
> >
> > > “Could you provide quantitative evaluation metrics for the Mistral-7B relation extraction against a reference set, and how does its quality impact downstream VideoQA performance and robustness?”
> > >
> >
> > Here, we provide an evaluation and comparison of relation-extraction quality with Mistral-7B vs. the more recent Qwen3-7B model, using Qwen2.5-VL to assess if the extracted relations exist in 1.5 randomly sampled videos from the Webvid dataset.
> >
> > | Extractor | Accuracy |
> > | --- | --- |
> > | Mistral-7B | 76.6 |
> > | Qwen3-7B | 82.1 |
> >
> > While we get good accuracy with Mistral-7B, the more recent LLM can further improve relation quality.

---

> > > ### Comment · Reviewer_NhbL · 2025-11-27
> > >
> > > Thank you to the authors for the clear and well-prepared rebuttal. The additional explanations and experiments improve the clarity of the method and address several of the earlier questions.
> > >
> > > After reading the response and considering the discussion with other reviewers, I will keep my original score. No further comments from my side.

---

### Official Review · Reviewer_SfZK · 2025-10-31

**Soundness:** 2
**Presentation:** 3
**Contribution:** 2
**Rating:** 4
**Confidence:** 4

**Summary:**

This paper introduces REVEAL, a relation-based video representation learning framework for VideoQA. The method encodes videos as sets of (subject, predicate, object) relation triplets extracted from captions via LLMs, then aligns visual “relation queries” (from a Q-Former) with text-based relation embeddings using a new Many-to-Many Noise Contrastive Estimation (MM-NCE) loss and Hungarian matching. The model is pretrained on WebVid-2M and evaluated on five VideoQA datasets (STAR, NExT-QA, Intent-QA, TVQA, VLEP), showing competitive or superior performance compared to recent VLM-based approaches.

**Strengths:**

-	The experimental evaluation is comprehensive, covering multiple challenging VideoQA datasets.

-	Relation understanding is a meaningful direction for improving compositional and semantic understanding in video-language learning.

**Weaknesses:**

-	The method assumes that vision queries and text-based relation embeddings are already semantically aligned (by choosing positive pair with Hungarian matching and cosine similarity), relying on representations. The MM-NCE loss primarily refines this existing bias rather than learning a new alignment.

-	The representation learning objective focuses solely on static object-level relations, ignoring temporal and causal relationships between events or entities. It is unclear how REVEAL achieves strong performance on causal or temporal reasoning benchmarks (e.g., Intent-QA, NExT-QA) compared to methods like Vamos, which explicitly incorporate temporal reasoning using the same backbone.

**Questions:**

-	Can you please explain why the method relies on the bias of pretrained models at the beginning and then refines it through contrastive learning? What happens if the relation understanding in those pretrained models is incorrect?
-	Is an objective that focuses only on object–object relation modeling truly effective for video understanding? How does this approach help the model improve on temporal or causal questions?

---

> ### Author Response · Authors · 2025-11-25
> **Response to Reviewer SfZK**
>
> We thank the reviewer for the constructive comments. We address both main concerns below.
>
> ---
>
> ### W1 and Q1. On “relying on pretrained bias” vs. learning alignment with MM-NCE
>
> > “The method assumes that vision queries and text-based relation embeddings are already semantically aligned … The MM-NCE loss primarily refines this existing bias rather than learning a new alignment.”
> >
> >
> > *“What happens if the relation understanding in those pretrained models is incorrect?”*
> >
>
> We would like to clarify that our method does not assume a pre-aligned video–text space. Only the **text side** optionnaly starts from a pretrained encoder; the **vision queries are randomly initialized** and acquire semantics entirely through training. At each optimization step, Hungarian matching is recomputed based on the *current* similarities, and MM-NCE then jointly updates the queries and, in our best-performing setting, the relation encoder itself, which is trainable and allowed to deviate from its initial Sentence-RoBERTa. Moreover, unmatched queries are not forced toward any text relation but still participate as negatives across the batch, so the alignment comes from the learned interaction between learnable queries, adapted text embeddings, and the contrastive signal, rather than from a static pretrained alignment. We have demonstrated this through two experiments:
>
> First:
>
> - **MM-NCE improves over simple matching even with a frozen encoder.**
> In Table 6.b:
>     - **Frozen + MSE loss** (Hungarian matching + regression),
>     - **Frozen + MM-NCE loss**, and
>     - **Trainable + MM-NCE loss**.
>
> | Relation Encoder + Loss | STAR All | NExT-QA All | Intent-QA All |
> | --- | --- | --- | --- |
> | Frozen + MSE loss | 66.4 | 71.1 | 68.9 |
> | Frozen + MM-NCE loss | 67.9 | 72.6 | 71.4 |
> | Trainable + MM-NCE loss | 69.4 | 74.0 | 71.8 |
>
> Across STAR, NExT-QA, and Intent-QA, we observe:
>
> - `Frozen + MSE` < `Frozen + MM-NCE` < `Trainable + MM-NCE`.
> This shows that:
> 1. Even when the relation encoder is frozen, **MM-NCE learns a better alignment** than simple MSE on matched pairs.
> 2. Allowing the relation encoder to be **trainable with MM-NCE** further improves performance, demonstrating that the model is not just “refining a fixed bias” but actively adapting the text embedding space to video relations.
>
> Secondly:
>
> - **The method does not require a strong initial text prior.**
> Table 7.a ablates the initialization of the relation encoder:
>     - Random init, RoBERTa-large, CLIP text encoder, and a contrastive sentence embedder all lead to strong downstream results, with the contrastive sentence embedder performing best but **random and non-contrastive inits still reaching competitive performance**.
>     This indicates that the model can learn meaningful alignment even without a text-encoder prior.
> - We hypothesize that with larger-scale pretraining, training the relation encoder from scratch could outperform initialization from a sentence embedder.
>
> | Relation Encoder Init | STAR All | NExT-QA All | Intent-QA All |
> | --- | --- | --- | --- |
> | Random init | 68.3 | 71.0 | 69.2 |
> | RoBERTa-large | 68.0 | 72.2 | 71.0 |
> | CLIP text encoder | 68.5 | 72.3 | 71.4 |
> | Sentence embedder | 69.4 | 74.0 | 71.8 |
> - **Robustness to extractor choice.**
>
> We evaluated relation extraction quality using a **VLM-as-a-judge** setup:
>
> We randomly sampled **1.5k caption–relation pairs** from the WebVid-2M–based pool of extracted relations.
> For each caption, we considered the triplets produced by **Mistral-7B** and by **Qwen3-7B** as two alternative candidates.
> We used **Qwen2.5-VL** as an automatic judge. For each (video, extracted triplet) pair, we prompted Qwen2.5-VL to decide whether the triplet is consistent with the video.
> The resulting numbers are:
>
> | Extractor | Accuracy |
> | --- | --- |
> | Mistral-7B | 76.6 |
> | Qwen3-7B | 82.1 |
>
> These results indicate that Mistral-7B already provides reasonably accurate relations, and that using a stronger extractor can further improve relation quality.

---

> > ### Author Response · Authors · 2025-11-25
> > **Response to Reviewer SfZK (Part 2)**
> >
> > ### W2 and Q2. On “static relations” vs. temporal and causal reasoning
> >
> > > “The representation learning objective focuses solely on static object-level relations, ignoring temporal and causal relationships … It is unclear how REVEAL achieves strong performance on causal or temporal reasoning benchmarks (e.g., Intent-QA, NExT-QA).”
> > >
> > >
> > > *“Is an objective that focuses only on object–object relation modeling truly effective for video understanding? How does this approach help the model improve on temporal or causal questions?”*
> > >
> >
> > Our relations are **not restricted to static object–object pairs** and are computed over **temporal sequences** with explicit temporal modeling:
> >
> > First, we argue that predicates encode actions and events, not just static links.
> > The extracted triplets from Webvid captions contain verbs as highlighted by the examples presented in Table 9 in the supplement. Furthermore, we also augment our data pool with video-relation pairs from the Charades dataset, which contains a significant amount of actions. These allow for capturing dynamic events rather than purely static configurations. Furthermore, Figure 4 in the supplement shows examples of our method successfully encoding these dynamic events: we plot the relation queries alignment from the encoded video segments, and for text relations derived from the questions and the different available choices. We can see that the relation derived from the question is the most aligned with the video segment in which this relation occurs. The same goes for the ground-truth answer. We also see that the negative choice relations are never better aligned than the correct answer relation.
> >
> > Here, we clarify how we perform temporal modeling in the proposed pretrained backbone:
> >
> > - A **Slow-Fast dual-pathway architecture**, where the Fast pathway processes CLS tokens across 16 frames and the Slow pathway processes patch features from 4 frames.
> > - **Dedicated temporal encoders** for each pathway that model dependencies across frames.
> > - Relation Q-formers that attend over these temporally encoded features to produce relation queries.
> >
> >     Thus, each relation query is derived from a **multi-frame temporal context**, not a single static snapshot, and temporal modeling occurs during pretraining when vision queries learn to attend to the correct frames in the Q-formers.
> >
> > - There is also further temporal modeling during VideoQA finetuning. We argue that it is a better choice to leverage the sequential nature of LLM to improve temporal modeling:
> >
> >     During VideoQA fine-tuning:
> >
> >     - Videos are segmented into multiple **temporal chunks**.
> >     - For each segment, 16 vision queries are produced and projected into the LLM space.
> >     - We add **learnable temporal tokens** that encode segment positions, and special tokens distinguishing Slow vs. Fast pathways.
> >     This provides the LLM with **temporally indexed relation tokens**, allowing it to reason over event order and causality.
> > - Experimentally, we observe particularly large gains on temporal splits over baselines.
> > - Finally, we have empirical evidence on the importance of temporal resolution: Supplementary Table 10.a shows that **temporal resolution** (number of segments) has a dataset-dependent optimum, and changing it affects performance. For instance, STAR significantly benefits from increased temporal resolution.
> >
> > | Temporal Resolution (# segments) | STAR All | NExT-QA All | Intent-QA All |
> > | --- | --- | --- | --- |
> > | 1 | 61.8 | 73.4 | 70.7 |
> > | 2 | 65.1 | 73.3 | 71.8 |
> > | 4 | 66.7 | 72.6 | 71.1 |
> > | 8 | 69.4 | 72.7 | 70.8 |
> >
> > This shows that, *e.g.*, STAR benefits notably from higher temporal resolution (61.8 → 69.4), while the optimal number of segments differs slightly across datasets, supporting your statement.
> >
> > In summary, although our supervision is expressed as subject–predicate–object triplets, the predicates capture actions, and these triplets are learned over temporally encoded video features and temporally segmented inputs. The resulting relation queries are therefore sequence-aware and provide the LLM with structured, temporally grounded tokens, which empirically support improvements on causal and temporal reasoning benchmarks.

---

### Official Review · Reviewer_E4mr · 2025-11-01

**Soundness:** 3
**Presentation:** 3
**Contribution:** 3
**Rating:** 6
**Confidence:** 5

**Summary:**

This paper proposes REVEAL, a framework that models videos as (subject-predicate-object) relation triplets. REVEAL extracts relations from captions via Mistral-7B, uses a Q-Former to generate video-derived visual queries, and introduces MM-NCE loss to align these queries with text-based relation embeddings. It adopts a Slow-Fast dual pathway and is evaluated on 5 benchmarks, at least competing with state-of-the-art models.

**Strengths:**

Overall, this paper is well-written, with clear motivation for the research problem and systematically designed experiments.

It effectively redefines video representation for VideoQA by modeling video content as (subject-predicate-object) relation triplets, moving beyond the limitations of traditional global alignment methods or closed-vocabulary scene graphs.

It proposes the MM-NCE loss to align unordered, incomplete sets of visual queries and text-based relation embeddings—successfully addressing the challenge of variable relation counts per video, a gap that prior losses (such as MIL-NCE) failed to cover.

Additionally, it adapts the Slow-Fast architecture and Q-Former for relation modeling (using the fast pathway for temporal aggregation and the slow pathway for fine-grained spatial details), forming a novel and practical combination.

**Weaknesses:**

1. The paper compares REVEAL to models like ViLA and VideoChat but omits post-2024 VLMs such as Qwen2.5-VL. This gaps makes its "competitive against SOTA" claims less convincing for current research.

2. The paper uses unique hyperparameters for each dataset, unlike baselines that often use unified settings. This could inflate REVEAL’s gains , as improvements might stem from tuning rather than its core design.

3. Given the paper currently only evaluates REVEAL on multiple-choice VideoQA benchmarks, its arguments would be more convincing if the authors considered involving more recent or challenging benchmarks, such as free-ended QA benchmarks like MSVD and MSRVTT.

**Questions:**

See weaknesses.

---

> ### Author Response · Authors · 2025-11-25
> **Response to Reviewer E4mr**
>
> We thank the reviewer for the positive assessment of our work and for the constructive suggestions. We are encouraged that the reviewer found the paper well-written with a clear motivation and extensive experimental design, and that our reformulation of video representation for VideoQA in terms of (subject–predicate–object) triplets is highlighted as a meaningful step beyond global alignment and closed-vocabulary scene graphs. We also appreciate the recognition of MM-NCE as a suitable way to align unordered, potentially incomplete sets of visual queries and text relations, a setting that traditional objectives like MIL-NCE do not directly handle, and of our use of the Slow–Fast architecture and Q-Former as a practical, effective combination for capturing both temporal context and fine-grained spatial details.
>
> ---
>
> ### W1 and W2. Broader coverage of recent VLMs and Dataset-specific hyperparameters and potential tuning bias
>
> > “The paper compares REVEAL to models like ViLA and VideoChat but omits post-2024 VLMs such as Qwen2.5-VL.”
> >
>
> Our main goal is to make **apples-to-apples** comparisons under matched backbones and training regimes.
>
> - **Apples-to-apples comparisons**: Across STAR, NExT-QA, Intent-QA, TVQA, and VLEP, we directly compare our results with those of Flipped-VQA and Vamos, using the same vision backbone (ViT-L/14) and LLM family (Llama1/Llama3), as shown in Tables 1-5. Under these strictly comparable settings, REVEAL consistently improves over the corresponding baselines (e.g., +2.5% on STAR, +2.0% on NExT-QA, +3.3% on Intent-QA, +0.8% on TVQA, and +1.2–2.5% on VLEP).
>
> > “The paper uses unique hyperparameters for each dataset, unlike baselines that often use unified settings. This could inflate REVEAL’s gains…”
> >
> - We plug REVEAL into existing VideoQA pipelines, namely Flipped-VQA and Vamos. Following their experimental setup, we adopt dataset-appropriate training settings. This allows for strict comparability with these baselines and isolates the contribution of REVEAL on the observed gains. As stated in the previous point, results are consistent across benchmarks.
>
> ---
>
> ### W3. Multiple-choice vs. free-form VideoQA
>
> > “Given the paper currently only evaluates REVEAL on multiple-choice VideoQA benchmarks, its arguments would be more convincing if … including free-ended QA benchmarks such as MSVD and MSRVTT.”
> >
>
> In downstream applications, our focus is on challenging multiple-choice VideoQA benchmarks (STAR, NExT-QA, Intent-QA, TVQA, VLEP). This setting allows:
>
> - **Controlled comparison** against strong existing methods with the same backbone and LLM (Flipped-VQA, Vamos), isolating the effect of relation-based representations and MM-NCE.
> - **Easiness and robustness of evaluation**: MCQ has been the de facto Video-LLMs evaluation setup for the most recent VideoQA benchmarks [1,2]. It is easier and more robust as it does not need to rely on LLM-as-a-judge pipelines. This is one of the reasons we only do MCQ.
>
> In future work, REVEAL can be combined with large-scale instruction setups as the architecture is compatible with free-form QA.
>
> [1] Fu, Chaoyou, et al. "Video-mme: The first-ever comprehensive evaluation benchmark of multi-modal llms in video analysis." In CVPR. 2025.
>
> [2] Wang, Weihan, et al. "Lvbench: An extreme long video understanding benchmark." In ICCV. 2025.

---

### Author Response · Authors · 2025-11-25
**General Response to all Reviewers**

We thank all reviewers for their constructive feedback. We appreciate that Reviewers 1 and 3 found the paper well-motivated and clearly written, and highlighted our reformulation of video representation as (subject–predicate–object) triplets as a meaningful shift beyond global or closed-vocabulary scene-graph representations. We also appreciate that all reviewers recognized MM-NCE as a key technical component for aligning unordered and potentially incomplete sets of vision queries and text relations, and that several reviewers emphasized the extensiveness of our experimental evaluation.

In the detailed responses, we focus on the questions raised across reviews: (i) **novelty and relation-based vs. generic fine-grained or caption-based alignment** (reviewers 1, 3, 4), (ii) **MM-NCE’s alignment behavior, its dependence on pretrained components, and robustness to noisy or incomplete relations** (reviewer 2, 3, 4), (iii) **how relation-based pretraining supports temporal and causal reasoning rather than only static object–object relations** (Reviewer 2, 3), (iv) **experimental protocol and baselines**, including backbone-matched comparisons, SOTA coverage, multiple-choice vs. free-form QA, and per-dataset hyperparameters (Reviewer 1, 4), and (v) **relation extraction quality and potential biases** of the LLM-based extractor (Reviewer 2, 3).

Reviewer 1: **E4mr**

Reviewer 2: **SfZK**

Reviewer 3: **NhbL**

Reviewer 4: **UJzk**

---

> ### Author Response · Authors · 2025-12-03
> **Summary of the rebuttal phase**
>
> For the sake of clarity, we briefly summarize how we addressed the main concerns raised across all reviewers.
>
> ### 1. Novelty: Relations + MM-NCE vs. Caption-Based / “Fine-Grained” Alignment
>
> **Reviewer 1 & Reviewer 3** saw the relation-centric formulation as promising; **Reviewer 4** questioned whether this is “just additional pretraining/fine-grained alignment.”
> Our main contribution is a **set-level, open-vocabulary relation alignment objective (MM-NCE)** that operates on unordered, variable-size sets of relations and queries, rather than global captions or single embeddings. Under identical backbones and data, replacing global caption NCE with relation-based MM-NCE yields large gains (Table 6a):
>
> | Pretraining Setup       | STAR All | NExT-QA All | Intent-QA All |
> |-------------------------|----------|-------------|---------------|
> | Captions + NCE loss     | 31.5     | 56.3        | 61.2          |
> | Relations + MM-NCE loss | 65.4     | 72.8        | 70.8          |
>
> We plug this pretraining into existing Flipped-VQA/Vamos pipelines with the **same Llama + ViT-L/14 backbones** and **frozen video encoders**, and observe consistent gains across STAR, NExT-QA, Intent-QA, TVQA, and VLEP, isolating the contribution of the relation-based objective rather than architecture changes.
>
> ### 2. MM-NCE, Alignment “Correctness,” and Pretrained Bias
>
> **Reviewer 2 and Reviewer 4** asked whether MM-NCE simply refines a pre-aligned space and what happens if pretrained relation understanding is wrong.
>
> - **Random queries, optionally trainable text encoder.** Vision queries start from random initialization and only acquire semantics through training. The relation encoder can be frozen or trainable; in our best setting, it is trainable and allowed to deviate from its initial Sentence-RoBERTa space.
> - **Better than matching + MSE, even when frozen.** Table 6b shows that MM-NCE improves over matching+MSE with a frozen encoder, and a trainable encoder further helps:
>
> | Relation Encoder + Loss | STAR All | NExT-QA All | Intent-QA All |
> |-------------------------|----------|-------------|---------------|
> | Frozen + MSE loss       | 66.4     | 71.1        | 68.9          |
> | Frozen + MM-NCE loss    | 67.9     | 72.6        | 71.4          |
> | Trainable + MM-NCE loss | 69.4     | 74.0        | 71.8          |
>
> - **Robust to text-encoder initialization.** Table 7a shows that random, RoBERTa, CLIP, and sentence-embedder inits all yield strong performance; the contrastive sentence embedder is best but strong text priors are not required.
> - **Relation retrieval and query usage (new).** We added a relation retrieval analysis on STAR and NExT-QA. At the video level, we measure Top‑k accuracy: whether at least one ground-truth relation appears in the Top‑k predictions of at least one query:
>
> | k  | STAR | Next‑QA |
> |----|---------------------|------------------------|
> | 1  | 53.69               | 30.02                  |
> | 3  | 83.69               | 58.53                  |
> | 5  | 90.58               | 72.51                  |
>
> Per-query usage shows that multiple queries contribute Top‑5 hits (no single dominant slot), supporting the claim that MM-NCE learns meaningful, structured query–relation alignment.
>
> | Query Index | STAR | Next‑QA |
> |-------------|-----------------------------------------|-------------------------------------------|
> | 0           | 66.48%                                  | 37.16%                                    |
> | 1           | 63.87%                                  | 33.00%                                    |
> | 2           | 30.00%                                  | 16.69%                                    |
> | 3           | 33.25%                                  | 13.27%                                    |
> | 4           | 44.86%                                  | 41.26%                                    |
> | 5           | 51.93%                                  | 12.04%                                    |
> | 6           | 43.58%                                  | 30.54%                                    |
> | 7           | 54.79%                                  | 16.23%                                    |
>
> **We have updated the manuscript with these new elements of analysis (subsection 4.5 highlighted in blue with table 8 and figure 4). This improves the understanding of how our method works.**

---

> > ### Author Response · Authors · 2025-12-03
> > **Summary of the rebuttal phase 2**
> >
> > ### 3. Temporal / Causal Reasoning vs. “Static Relations”
> >
> > **Reviewer 2 & Reviewer 3** asked how our objective supports temporal/causal reasoning and whether we only model static object–object pairs.
> >
> > We clarified that:
> >
> > - Significant amounts of predicates in our triplets encode actions and events (e.g., “open door,” “sit at table”), notably the temporally annotated relations from Charades and VidOR.
> > - The backbone includes a **Slow–Fast dual-pathway** and **temporal encoders**, and relation queries attend over multi-frame features; temporal modeling is built into pretraining.
> > - During fine-tuning, we segment videos into **temporal chunks**, attach **temporal tokens** and Slow/Fast markers, and feed temporally indexed relation tokens into the LLM.
> >
> > Empirically, we see that (Supplementary Table 10.a) **temporal resolution** (number of chunks) matters, with STAR benefiting from higher resolution (61.8 → 69.4 as segments increase from 1 to 8).
> >
> > ### 4. Experimental Protocol, Baselines, and MCQ vs. Free-Form QA
> >
> > **Reviewer 1 & Reviewer 4** raised concerns about SOTA coverage, per-dataset hyperparameters, and the focus on multiple-choice QA.
> >
> > - **Backbone-matched baselines.** Our core claims are made against Flipped-VQA and Vamos with exactly the same ViT-L/14 and Llama backbones and similar finetuning regimes. Under these conditions, REVEAL consistently improves performance on all five benchmarks (e.g., +2.5% on STAR, +3.3% on Intent-QA).
> > - **General-purpose VLMs.** On NExT-QA we also report recent large VLMs (LLaVA-Next-Interleave, mPLUG-OWL-3, LLaVA-One Vision) and clearly annotate their much larger backbones and instruction-tuning data; we treat these as contextual references, not direct apples-to-apples baselines.
> > - **Hyperparameters.** We follow Flipped-VQA/Vamos in using dataset-specific hyperparameters (documented in the supplement) and keep the REVEAL video backbone frozen, which limits the degree to which tuning can explain the observed gains.
> > - **MCQ vs. free-form.** We focus on MCQ to allow controlled, comparable evaluation and avoid LLM-as-a-judge noise. The architecture remains fully compatible with free-form QA (relation tokens are simply projected into the LLM’s token space and can be used by a generative head); this is a natural direction for future work.
> >
> > ### 5. Relation Extraction Quality and Priors
> >
> > **Reviewer 2 & Reviewer 3** asked what happens if the LLM-based relation extractor is wrong or biased.
> >
> > We evaluated relation extraction using a **VLM-as-a-judge** protocol on 1.5k sampled examples:
> >
> > | Extractor  | Accuracy (Qwen2.5-VL as judge) |
> > |-----------|---------------------------------|
> > | Mistral-7B| 76.6                            |
> > | Qwen3-7B  | 82.1                            |
> >
> > Mistral-7B already provides reasonably accurate relations, and Qwen3-7B further improves quality.
> >
> > ### 6. Updates to the Manuscript
> >
> > Following requests from R4, we updated figures for better visibility and fixed citations in the revised version. We have also added an analysis of vision queries subsection in the experiments section, following new experiments and what has been discussed in point 2 of this reply.

---

### Meta-Review · Area_Chair_UYvR · 2026-01-07

**Summary:**

## Summary

The paper proposes REVEAL, a framework that models videos as (subject-predicate-object) relation triplets for Video-QA. The paper initially recieves a mixed rating of 6, 6, 4, and 2. After rebuttal, most of reviewers keep their ratings the same (both positive: i.e., Reviewer NhbL and negative: i.e., UJzk).

## Decision
AC reads all reviews and discussion and found that the concerns outweigh the merits of the paper at its current form. AC believes that the paper may be benefited from more revisions. Thus recommends to reject this paper at its current form and encourages the authors improve their paper and re-submit to future conferences.

**Reviewer Concerns:**

* The experiment setup and results of the paper are not conving (E4mr).
* Only evaluates REVEAL on multiple-choice VideoQA benchmarks (E4mr).
* Some strong assumption about alignment between text and vision (SfZK).
* Ignoring temporal and causal relationships between events or entities (SfZK).
* The method’s novelty is limited. Theoretical justification for MM-NCE and evidence of its alignment correctness are lacking. Experimental gains are modest, with missing comparisons to recent methods (UJzk).

**Reviewer Scores:**

The paper initially recieves scores of 6, 4, 6, 2 from four different reviews.

---

### Decision · Program_Chairs · 2026-01-26

Reject